# The *Acrasis kona* genome and developmental transcriptomes reveal deep origins of eukaryotic multicellular pathways

Sanea Sheikh [1,4,6], Cheng-Jie Fu[1,5,6], Matthew W. Brown[2,3] & Sandra L. Baldauf [1] ✉

Acrasids are amoebae with the capacity to form multicellular fruiting bodies in a process known as aggregative multicellularity (AGM). This makes acrasids the only known example of multicellularity among the earliest branches of eukaryotes (the former Excavata). Here, we report the *Acrasis kona* genome sequence plus transcriptomes from pre-, mid- and post-developmental stages. The genome is rich in novelty and genes with strong signatures of horizontal transfer, and multigene families encode nearly half of the amoeba's predicted proteome. Development in *A. kona* appears molecularly simple relative to the AGM model, *Dictyostelium discoideum*. However, the acrasid also differs from the dictyostelid in that it does not appear to be starving during development. Instead, developing *A. kona* appears to be very metabolically active, does not induce autophagy and does not up-regulate its proteasomal genes. Together, these observations strongly suggest that starvation is not essential for AGM development. Nonetheless, development in the two amoebae appears to employ remarkably similar pathways for signaling, motility and, potentially, construction of an extracellular matrix surrounding the developing cell mass. Much of this similarity is also shared with animal development, suggesting that much of the basic tool kit for multicellular development arose early in eukaryote evolution.

Eukaryotes employ two basic modes of multicellularity. In clonal multicellularity, a single cell develops into a multicellular organism by coordinated growth and differentiation. In aggregative multicellularity (AGM), growth and differentiation occur separately, intersected by a striking transition from solitary (asocial) growth to social development. Both multicellular strategies require cell-cell signaling, interaction, and cooperation, and both have evolved multiple times. However, while clonal multicellularity is well studied in several very different systems, e.g., plants and animals, AGM has only been extensively studied in the dictyostelid amoebae, especially *Dictyostelium discoideum*,

including hundreds of researchers worldwide and extensive published, cultured and online resources such as the multilayered community resource DictyBase (dictybase.org). We are studying AGM in the heterolobosean amoeba, *Acrasis kona*, which is separated from dictyostelids by over a billion years of evolution[1].

Acrasids are large, fast-moving amoebae, common in the wild, and easily grown in the lab (Supplementary Movie 1)[2]. After several days on solid media, amoebae enter a social stage with groups of cells aggregating in regions devoid of food. The resulting aggregates surround themselves with an extracellular slime sheath and then commence to

[1]Program in Systematic Biology, Department of Organismal Biology, Uppsala University, Uppsala, Sweden. [2]Department of Biological Sciences, Mississippi State University, Mississippi State, Mississippi, USA. [3]Institute for Genomics, Biocomputing & Biotechnology, Mississippi State University, Mississippi State, MS, USA. [4]Present address: Section of Terrestrial Ecology, Department of Biology, University of Copenhagen, Copenhagen, Denmark. [5]Present address: Olink, Division of Thermo Fisher Scientific, Uppsala, Sweden. [6]These authors contributed equally: Sanea Sheikh, Cheng-Jie Fu. ✉e-mail: sandra.baldauf@ebc.uu.se

develop into a multicellular fruiting body (sorocarp, Fig. 1). Morphogenesis begins with the basal sorogen cells encysting individually to form a stalk composed of irregularly-shaped thick-walled cysts. The stalk continues to grow as more basal cells are added, gradually lifting the remaining cell mass off the substrate. Once the stalk is complete, the aerial cell mass proceeds to form lobes, which then elongate to form branching uniseriate rows of cells. Once the aerial array assumes its final configuration, the cells proceed to encyst *en mass* to form uniformly rounded, thick-walled spores. Encysted aerial cells (spores) are further differentiated from encysted stalk cells by the presence of raised, highly pigmented, ring-like hila (areolae) at each spore-spore contact point[3]. Four *Acrasis* species have been described, each with a distinct sorocarp morphology, ranging from a simple stalk to the multiply branching tree-like structures of *A. kona* (Fig. 1)[2].

The acrasid life cycle is strikingly similar to that of the dictyostelids[4], and acrasids were long considered their primitive relatives[5]. Due at least in part to this presumed close relationship, *Acrasis rosea* (now *Acrasis kona*) enjoyed some popularity as an experimental model, especially for the study of cytoskeletal and related features (e.g.,)[6]. However, the amoebae of acrasids and dictyostelids differ markedly in both morphology and behavior, suggesting that they may be only distant relatives[3,7], and molecular phylogeny now places them widely separated in the eukaryote tree[8]. Thus, acrasids are now placed together with the model organism *Naegleria gruberi* in phylum Heterolobosea (suprakingdom Discoba)[9], while dictyostelids are placed with animals and fungi in suprakingdom Amorphea[10]. We are studying the evolution of development using *A. kona* as a model system, beginning with the sequencing of its genome and three life-cycle stage-specific transcriptomes (Fig. 1). This allows us to model the critical transition from asocial feeding to social development in *A. kona* and compare it to the corresponding transition in the dictyostelid model, *Dictyostelium discoideum* AX4 (Ddi AX4)[11]. Our results reveal a remarkable similarity in

central developmental pathways shared by the acrasid and dictyostelid and, in some cases, even animals.

## Results

### The Acrasis kona genome

We estimate the *A. kona* genome to be 44.02 Mbp in size, based on the genome assembly (Supplementary Tables S1, S2), and essentially complete, based on its annotation (92.7% CEGMA, 93.1% BUSCO; Supplementary Table S3)[12,13]. Roughly half of the *A. kona* genes are predicted to have introns, with an average of 1-2 introns per intron-containing gene and a size range of 9–95 bp, although there are 54 predicted genes with 10 or more introns (maximum 21, AKO1_015466). Altogether 28% of the genes have predicted transmembrane domains (TMs) and/or signal peptides (SPs) (Supplementary Data 1). The genome appears to be rich in novelty, with nearly a third of the 15868 predicted proteins having no GenBank BLASTp hits below e-5 (5987 accessions). Repeats are also numerous and diverse, covering 16.8% of the genome compared to 5.1% of the genome of *Naegleria gruberi* (Supplementary Table S4), one of the few heterolobosean taxa with a well-characterized genome[14]. The *A. kona* predicted proteome is also highly redundant, with nearly half of the predicted proteins clustering into 2728 orthology groups (OGs) ranging in size from 2 to 49 members (Supplementary Table S5 and Supplementary Data 2). Most of these families seem to have evolved relatively recently as the vast majority are single-copy or absent in *N. gruberi* (2368 families), and over half are single-copy or absent across a wide sampling of eukaryotes (1607 families, Supplementary Data 2). *A. kona* membrane proteins are enriched for protein families: 63% of proteins with four or more transmembrane domains are assigned to orthology groups (1225 proteins, Supplementary Data 1). *A. kona* is also rich in genes apparently acquired by horizontal gene transfer (HGT), and both multicopy and, especially HGT genes are enriched for metabolic functions and depleted for functions involved in information processing

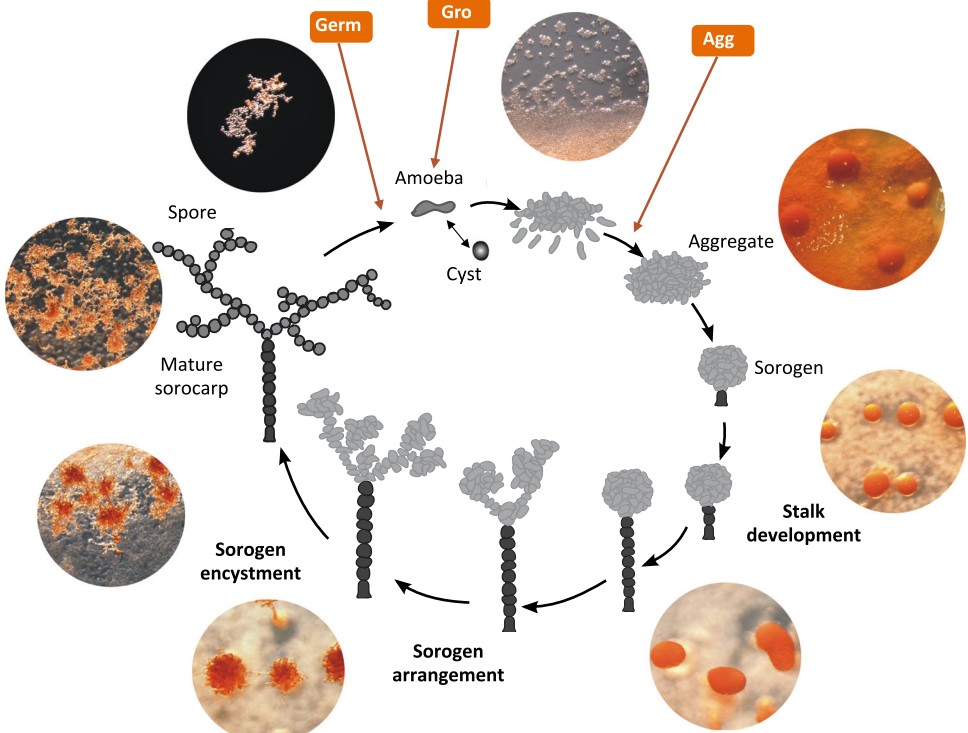

**Fig. 1 | The *Acrasis kona* life cycle.** The acrasid life cycle alternates between a single-celled (asocial) feeding stage and a multicellular (social) dispersal stage. Development begins with aggregation, where cells migrate together to form a ball of cells (sorogen) surrounded by an extracellular matrix (slime sheath). The basal cells then encyst to form a stalk, gradually lifting the remaining cell mass above the substrate. With the stalk complete, the aerial cells align into chains and then encyst *en mass*, creating a mature multicellular fruiting body (sorocarp)[3]. Time points for the three life cycle stage transcriptomes are indicated with orange arrows and labeled as follows: Germ (spore germination), Gro (growth), and Agg (aggregation).

**Table 1 | Comparative distribution among general metabolic function categories for *Acrasis kona* single-copy, multi-copy, and horizontally transferred genes**

| Expasy category | multi-copy | | single-copy | | HGT | | Category description |
|---|---|---|---|---|---|---|---|
| | accs | % ttl | accs | % ttl | accs | % ttl | |
| **Metanbolism** | | | | | | | |
| Carbohydrates | 33 | 8.0% | 167 | 7.5% | 59 | 12.1% | carbohydrate metabolism |
| Energy | 19 | 4.6% | 80 | 3.6% | 15 | 3.1% | energy metabolism |
| Lipids | 30 | 7.3% | 113 | 5.1% | 34 | 7.0% | lipid metabolism |
| Nucleotides | 12 | 2.9% | 63 | 2.8% | 19 | 3.9% | nucleotide metabolism |
| Amino Acids | 39 | 9.5% | 168 | 7.5% | 44 | 9.1% | amino acid metabolism (all) |
| Glycans | 7 | 1.7% | 78 | 3.5% | 15 | 3.1% | glycan biosynthesis and metabolism |
| Cofactors | 15 | 3.7% | 85 | 3.8% | 17 | 3.5% | metabolism of cofactors and vitamins |
| other | 11 | 2.7% | 81 | 3.6% | 32 | 6.6% | secondary metabolites |
| **total** | **166** | **40.4%** | **835** | **37.4%** | **235** | **48.4%** | |
| **Genetic Information Processing** | | | | | | | |
| Proteins | **77** | **18.8%** | **581** | **26.0%** | **60** | **12.3%** | transcription, translation, folding, sorting, degradation |
| **Environmental Information Processing** | | | | | | | |
| Signaling | **54** | **13.2%** | **264** | **11.8%** | **64** | **13.2%** | membrane transport, signal transduction, molecules, interaction |
| **Cellular Processes** | | | | | | | |
| Transport | **35** | **8.5%** | **152** | **6.8%** | **46** | **9.5%** | transport and catabolism |
| Cell Cycle | **50** | **12.2%** | **291** | **13.0%** | **48** | **9.9%** | replication and repair, cell growth, death, auto/mitophagy |
| Cell-cell interact | **14** | **3.4%** | **74** | **3.3%** | **21** | **4.3%** | cellular community – eukaryotes, prokaryotes |
| Motility | **14** | **3.4%** | **37** | **1.7%** | **12** | **2.5%** | cell motility |

*A. kona* predicted proteins were assigned to functional categories using BLASTKoala[87], and a number of accessions (accs) and percentages of subcategories (% of ttl) were calculated after the removal of redundancies. Numbers in bold font indicate totals for the respective category.

Major categories and their respective total numbers of accessions are indicated in bold font.

(Table 1). The *A. kona* metabolic repertoire is diverse (Supplementary Tables S6, Supplementary Fig. S1, and Supplementary Data 3) except for lacking anaerobic capacity, which is indicated by the absence of Fe-hydrogenase and pyruvate:ferredoxin oxidoreductase (Supplementary Data 3).

Phylogenetic analysis of all taxonomically widespread *Acrasis kona* HGT candidates (multiHits) yields 1253 HGTs with $\geq 60\%$ maximum likelihood ultrafast bootstrapping support (HGT$_{\geq 60}$, Fig. 2). HGT uniHits, accessions found in only one major non-excavate taxon, are probably also largely legitimate, although they cannot be independently tested with phylogeny due to the lack of an outgroup. For example, *A. kona* uniHits found in diverse Bacteria are almost certainly true bacteria-to-eukaryote transfers (e.g., Fig. 3A). Many of these bacterial uniHits also have introns (e.g., Fig. 3B) and signal peptides (e.g., Fig. 3C) in *A. kona*, confirming that the sequences are not bacterial contaminants and have a specific function in the amoeba. Over half of the HGT$_{\geq 60}$ accessions belong to orthology groups (692 accessions). These HGT families are often monophyletic, indicating post-transfer expansion (e.g., Fig. 3A). However, the bulk of redundant *A. kona* HGT$_{\geq 60}$ accessions are not monophyletic (496 accessions). This signifies either multiple independent transfers or, more often, a combination of horizontal and vertical transmission, in which HGTs cluster with non-HGT accessions (mixed mfams) (Fig. 2A). Thus, while HGT is notorious as a source of major genetic novelty, the majority of *A. kona* HGTs appear to be redundant, although possibly mildly expanding the metabolic capacity of this omnivorous micro-predator (Supplementary movie 1)[3]. The acrasid HGT$_{\geq 60}$ accessions also trace to all domains of life but especially to the taxon groups Ciliophora, Plasmodiophora, Oomycota, and Fungi (Fig. 2B, C). These four taxa are largely soil microbes, including species probably abundant on the dead or dying vegetation where acrasids are almost exclusively found[2,3], suggesting that acrasids mostly acquire HGTs from their prey.

Intra- and extra-cellular signaling is critical for all organisms, including microbes. Social microbes have the added requirements of attracting aggregation partners, directing their movement in the aggregate, and organizing them into a fruiting body. *Acrasis kona* encodes a wealth of signaling domains, especially cyclase and calcium-binding (EF-hand) domains, RAS GTPases, and hetero-trimeric G-protein regulators (Fig. 4 and Supplementary Data 4). The acrasid genome is also uniquely, if mildly enriched in nearly all components of phosphatidylinositol (PIP) signaling and blue light sensor (BLUF) domains, the latter also a powerful inducer of *A. kona* development[3]. Hybrid histidine kinases (hybrid hisK) stand out especially as far more abundant in *A. kona* than any of a diverse sampling of eukaryotes (Fig. 4). For example, *A. kona* has more than twice the number of hisK kinase (HK) and response regulator (RRR) domains (60 and 83, respectively) than its closest examined relative, *Naegleria gruberi* (28 and 33, respectively; Supplementary Data 4). The acrasid also has a wealth of proteins with PAS domains, which are common hisKR interactors (IPR013767).

**Developmental gene expression in Acrasis kona**

To gain insight into *Acrasis kona* development, we sequenced transcriptomes from pre-, mid-, and post-developmental cells, which correspond to the primary life cycle stages of growth (Gro), aggregation (Agg), and spore germination (Germ), respectively (Fig. 1). We focused particularly on genes associated with development by comparing gene expression between Gro and Agg, i.e., actively growing versus aggregating cells. Genes with substantially increased or decreased expression in aggregation over growth are referred to as Aggup and Aggdn, respectively. For comparison, we also analyzed Agg versus Germ to identify genes with increased or decreased expression in germination over aggregation, i.e., return to active growth (Germup and Germdn, respectively) (Supplementary Data 5). Substantial differential expression (SDE) was defined based on a combination of length-corrected read

**A**

| | A. kona and N. gruberi Non-excavate Top Hits - General Properties | | | | | | | | | | | |
|---|---|---|---|---|---|---|---|---|---|---|---|---|
| | *A. kona* all non-excavate | | *A. kona* non-excavate multiHits with ≥60% bootstrap support (HGT≥60) | | | | | | | | *N. gruberi* all non-excavate top hits | |
| | | | all HGT≥60's | | | | | multicopy HGT≥60's only | | | | |
| | multiHit | uniHit | number | % of total | TM | SP | TM + SP | families | accessions | mono fams | number | % of total |
| Amoebozoa | 916 | 71 | 201 | 16.0% | 41 | 23 | 2 | 90 | 121 | 10 (20 acc's) | 155 | 16.6% |
| Metazoa | 836 | 46 | 162 | 12.9% | 30 | 28 | 12 | 68 | 88 | 7 (15 acc's) | 149 | 16.0% |
| Fungi | 697 | 80 | 345 | 27.5% | 49 | 49 | 4 | 109 | 195 | 44 (95 acc's) | 110 | 11.8% |
| Archaeplastida | 675 | 39 | 121 | 9.7% | 26 | 20 | 4 | 47 | 64 | 6 (13 acc's) | 101 | 10.8% |
| SAR | 645 | 43 | 263 | 21.0% | 49 | 40 | 10 | 77 | 132 | 10 (22 acc's) | 260 | 27.9% |
| Bacteria | 486 | 52 | 95 | 7.6% | 15 | 21 | 4 | 35 | 55 | 9 (21 acc's) | 93 | 10.0% |
| Archaea | 130 | 10 | 66 | 5.3% | 3 | 5 | 0 | 29 | 37 | 5 (10 acc's) | 64 | 6.9% |
| total | 4385 | 341 | 1253 | | 213 | 86 | | 455 | 692 | | 932 | |

**B**

| A. kona multiHit HGT≥60 taxonomy - single HGT events | | | | | | | | | |
|---|---|---|---|---|---|---|---|---|---|
| Single events: single copy HGTs and monophyletic HGT families (post-HGT gene family expansions) | | | | | | | | | |
| SAR: | | | | | Fungi: | | | | |
| taxnomy | | HGT≥60 accessions | mono familiess | total HGT events | taxnomy | | HGT≥60 accessions | mono families | total HGT events |
| Alveolates | Apicomplexa | 16 | | 16 | Ascomycota | Leotiomyceta | 158 | 16 | 142 |
| | Ciliophora | 104 | 6 | 99 | | Saccharomycetes | 26 | 3 | 23 |
| | Vitrella spp. | 14 | 2 | 12 | | Schizosaccharomycetales | 6 | | 6 |
| | Perkinsus spp. | 6 | | 6 | | misc (2 families) | 8 | | 8 |
| | misc | 3 | | 3 | Basidiomycota | Agaricales | 13 | 2 | 11 |
| Rhizaria | Plasmodiophora | 30 | | 30 | | Polyporales | 12 | 1 | 11 |
| | Reticulomyxa spp. | 15 | 2 | 13 | | Boletales | 10 | 1 | 9 |
| Stramenopiles | Blastocystis spp. | 4 | | 4 | | Russulales | 10 | 1 | 9 |
| | Bacillariophyta | 16 | | 16 | | misc (10 orders) | 32 | 2 | 30 |
| | Nannochloropsis | 3 | | 3 | Chytridiomycota | Spizellomyces | 44 | 8 | 36 |
| | Aureococcus spp. | 3 | | 3 | | Batrachochytrium | 9 | | 9 |
| | Ectocarpus spp. | 2 | | 2 | Microsporidia | | 5 | | 5 |
| | Oomycetes | 47 | | 47 | Mucoromycota | | 8 | | 8 |

**C**

| A. kona multiHit HGT≥60 taxonomy - multiple events | | | |
|---|---|---|---|
| Multiple events: non-monophyletic HGT families (multiple transfer events or mixed horizontal and vertical transmission) | | | |
| SAR: | | Fungi: | |
| multiple SAR donors | 1 | multiple fungal donors | 9 |
| 1 SAR donor + ≥1 non-SAR donor | 83 | 1 fungal donor + ≥1 non-fungal donor | 52 |
| multiple SAR donors + ≥1 non-SAR donor | 4 | multiple fungal donors + ≥1 non-fungal donor | 14 |

**Fig. 2 | Patterns of horizontal gene transfer (HGT) in *Acrasis kona*. A** Total numbers of predicted proteins (accession or accs) with non-excavate top hits (BLASTp) are shown and organized by top hit taxonomy for *A. kona* (left side, blue background) and *Naegleria gruberi* (far right, green background). *A. kona* hits are further classified as present in one (uniHits) or multiple (multiHits) non-excavate taxon groups. *A. kona* multiHits with >60% bootstrap support (HGT≥60) were further screened for signal peptides (SP), transmembrane domains (TM), and orthology group membership (families or fams). HGT≥60 families are either monophyletic (mono fams), indicating a single transfer followed by duplication (*e.g.*, Supplementary Fig. S3), or mixed non-monophyletic families (*e.g.*, Supplementary Fig. S4ad), indicating multiple independent transfers or a mixture of horizontal and vertical transmission (*e.g.*, Supplementary Fig. S4af). Panels **B** and **C** show the top hit taxonomy of multiHit HGTs arising from single (**B**) or multiple (**C**) transfer events.

numbers (RPKM) and fold-change in expression (DE$_{Log2}$, Supplementary Data 6). Since we have only a single replicate for each of the three *A. kona* life cycle stages, *A. kona* DE$_{Log2}$ values were calculated using GFOLD (Supplementary Data 5)[15]. Gro vs Agg in *A. kona* appears roughly equivalent to growth versus five hours of starvation in Ddi AX4 (0_5h, mid-aggregation) in the dictyostelid (Supplementary Fig. S2)[11].

Aggregating *Acrasis kona* shows 448 accessions that are SDE Aggup, and many of these appear to be aggregation-specific as they show decreased expression at germination (Germdn, Supplementary Data 7). Altogether, 901 *A. kona* accessions show a strong response to aggregation (448 Aggup + 453 Aggdn, Table 2A) or less than 6% of the total predicted proteome. This is in sharp contrast to aggregating Ddi AX4, where 2595 accessions are Aggup and 1898 Aggdn, which together account for over 35% of its proteome (Table 2A)[16,17]. Thus, aggregation appears to be much simpler in *A. kona* relative to Ddi AX4 or even relative to *A. kona* germination (1188 Germup + 1641 Germdn; Table 2A). *A. kona* Aggup accessions also exhibit a level of novelty well below the genome average (22.3% vs 31.4%, respectively; Table 2A). Another form of novelty is the expansion of protein families. While the acrasid's Aggup accessions are no more likely to belong to orthology groups than the genome average (42.4% vs 43.0%, respectively; Table 2B), the Aggup-containing OGs are far less likely to be unique to *A. kona* (32.1%) compared to Germup (52.3%) or the genome as a whole (64.6%). Thus, aggregation in *A. kona* does not appear to have involved extensive

molecular innovation, particularly compared to germination, growth (Aggdn, 42.6% novel accessions), or the genome as a whole.

AGM is very widely if sporadically, distributed across eukaryotes[18]. One possible explanation for this could be the horizontal transfer of some critical factor, most likely originating from the only diverse and ancient AGM lineage, the Dictyostelia. To investigate this possibility, we constructed full molecular phylogenies for all *Acrasis kona* HGT≥60 Aggup accessions. Phylogenetic analysis supports HGT for 35 *A. kona* Aggup genes (Supplementary Data 8 and Supplementary Figs. S3, S4). A Pi-PLC of fungal origin is particularly interesting as it is a key component in phosphoinositide signaling and appears to be strongly aggregation-specific in *A. kona* (AKO1_013406, DE Aggup 2.79, DE Germdn − 3.93; Supplementary Data 7 and Supplementary Fig. S3, see below). However, only one of the *A. kona* Aggup HGT≥60 accessions appears to trace specifically to Dictyostelia (AKO1_002520, Supplementary Fig. S4f). AKO1_002520 appears to encode a 317 amino acid protein with a conserved transformer-2 (Tra2) domain, an RNA-binding motif involved in intron splicing control in human[19]. Although sequence similarity between the acrasid and dictyostelid accessions is strong (50% identity, 66% similarity), it is limited to the ~112 amino acid Tra2 domain. The Ddi AX4 homolog is also Aggdn (− 0.95 DE$_{1.5}$, Supplementary Data 7) and has no known developmental phenotype (DictyBase reference DDB_G0278349). The *Acrasis* sequence also branches as a distant sister lineage to a monophyletic Dictyostelia, including representatives of all

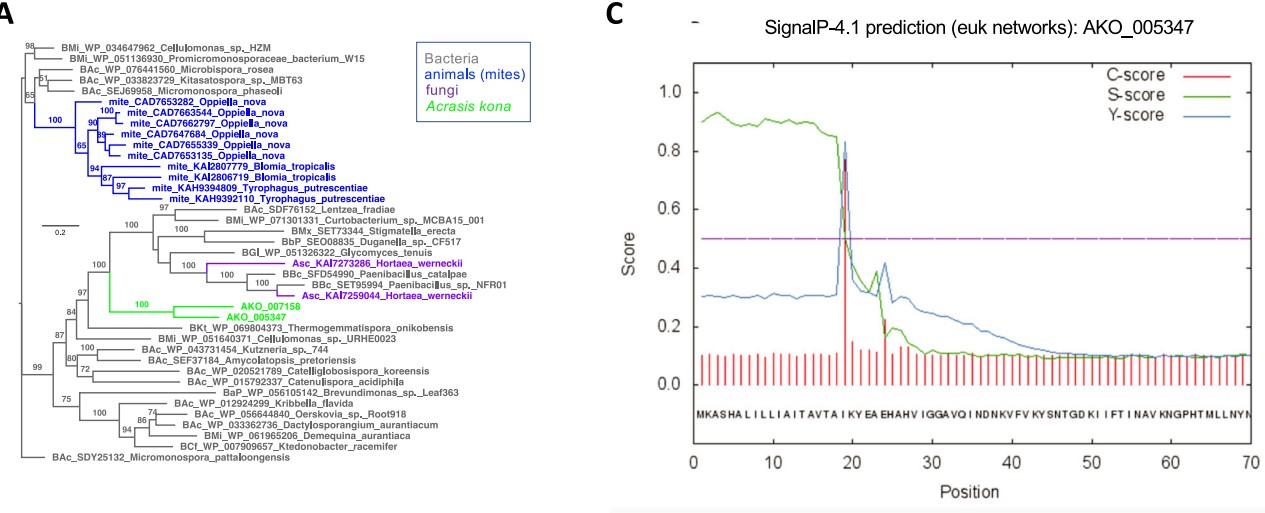

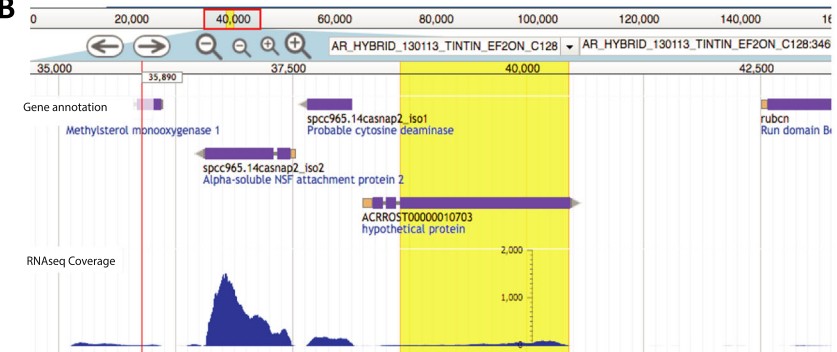

**Fig. 3 | Horizontal transfer of a bacterial gene into *Acrasis kona* included post-transfer addition of a signal peptide sequence and introns, followed by gene duplication.** Two copies of a sequence-conserved protein of unknown function are found in *A. kona* and a scattering of other eukaryotes trace to Bacteria, indicating gene acquisition by horizontal gene transfer (HGT). **A** HGT from Bacteria to *A. kona* followed by gene duplication is indicated by maximum likelihood analysis of a 503 amino acid alignment. **B** Both *A. kona* sequences carry a 5′ signal peptide sequence and two identically-placed introns (shown for AKO1_005347), and (**C**) with one intron lying at the junction between a predicted signal peptide sequence and the mature protein coding sequence (highlighted in yellow).

major divisions of the group (Supplementary Fig. S4f)[20]. This suggests that, if there was a gene transfer, it was either ancient or derived from a fairly distant relative of Dictyostelia.

### Development in *Acrasis kona* versus Dictyostelium discoideum

The switch from solitary feeding to social aggregation is considered the pivotal event in AGM development, and the Aggup repertoire is key to this shift. Nonetheless, only 448 accessions or less than 3% of the *A. kona* predicted proteome is Aggup. This contrasts with Ddi AX4, where as many as 2595 genes are Aggup (Table 2A), suggesting that aggregation is molecularly much simpler in the acrasid than in the dictyostelid. Even allowing for possible under-estimation by GFold of DE based on a single RNAseq replicate[15], necessitated by the availability for *A. kona* of only a single replicate per time point, the developing dictyostelid still shows substantially increased expression of at least 4-fold more genes than the acrasid (Table 2A). To investigate this further, we manually annotated the *A. kona* Aggup accessions using a consensus of their top BLASTp hits in GenBank nr, human Swissprot, and Ddi AX4 RefSeq, along with linked data in the conserved domain database (www.ncbi.nlm.nih.gov/cdd/), InterPro, and the extensively annotated dictyostelid database, DictyBase (dictybase.org[21]); The *A. kona* accessions were then clustered into broad functional categories based on their consensus annotation, and their expression compared with that of their homologs in Ddi AX4[11,22]. The protein function profiles of the two amoebae were then compared separately for active growth and feeding (0 h, Fig. 5A) and for aggregation (5 h, Fig. 5B).

There are 370 *Acrasis kona* Aggup accessions for which function could be assigned with confidence (Supplementary Data 7). The predicted proteins show a broad functional profile, including large components of all aspects of protein production (translation, protein modification/folding/sorting and degrading), as well as carbohydrate and lipid chemistry, cell cycle and nucleotide synthesis, and cellular signaling (Supplementary Data 7). During active growth (0 h), the numbers of these accessions that are actively expressed (RPKM > 10) show a very similar profile for *A. kona* accessions and their Ddi AX4 homologs, including similar numbers of accessions in nearly all functional categories (Fig. 5A). In fact, the two profiles are nearly identical, with the exception of extracellular/secreted/defense proteins, which are expected to be more species-specific given these two organisms' different preferred growth conditions and food sources.

However, after five hours without food, the Ddi AX4 and *A. kona* profiles diverge markedly (Fig. 5B). While Ddi AX4 shows massively reduced (Aggdn) expression of accessions involved in most aspects of information processing, including nearly all ribosomal proteins and tRNA synthetases, aggregating *A. kona* appears to be synthesizing proteins in a manner similar to actively growing cells (Fig. 5B). The acrasid's Aggup repertoire includes 14 ribosomal proteins, some of which are among the most highly expressed Aggup accessions, nine tRNA synthetases, five heat shock proteins and a nearly complete set of clathrin-coated vesicle subunits, some in multiple copies (Supplementary Data 7). Aggregating *A. kona* also seems to have very active mitochondria, with Aggup accessions for diverse mitochondrial

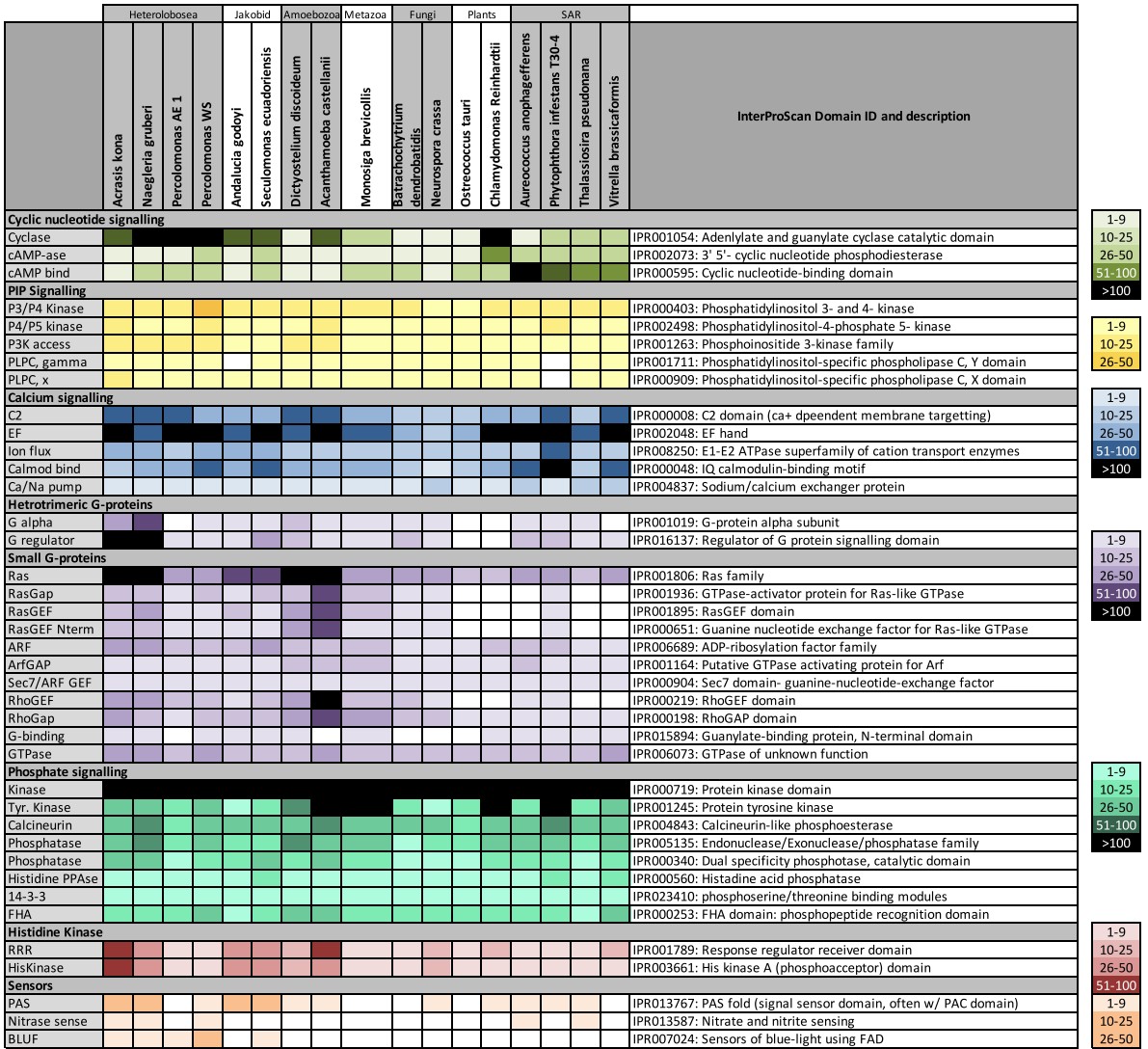

**Fig. 4 | Presence and relative abundance of common signaling domains in *Acrasis kona* and other eukaryotes.** Signaling domains on the left were identified by InterProScan[83] for *A. kona* and diverse other eukaryotes, including five additional representatives of kingdom Discoba: two Jakobida (*Seculamonas equadoriensis* and *Andalucia godoyi*) and three distantly related Heterolobosea (*Naegleria gruberi*, *Percolomonas* strain AE1 and *Percolomonas* strain WS). InterProScan IDs and descriptions of each domain are shown in the righthand column. Domain abundance is indicated by color intensity according to the keys at the far right, while exact numbers of domains for all taxa and categories are given in Supplementary Data 4. The list of domains is based on Fritz-Layland et al.[14].

functions, including protein synthesis, the TCA cycle, ATP synthesis/transport, and iron-sulfur cluster assembly (Supplementary Data 7). In contrast, aggregating Ddi AX4 shows a nearly complete loss of all aspects of mitochondrial maintenance and function, consistent with reports of macroautophagy in aggregating Ddi (see below[23,24]); Thus, five hours after the onset of aggregation, developing Ddi AX4 looks like a starving cell, while developing *A. kona* does not. In fact, there are only three functional categories in which the two amoebae show similar and substantial numbers of homologous Aggup accessions: signaling, protein folding/sorting, and cell shape and motility (cytoskeleton; Fig. 5B). Thus, in general, aggregating *A. kona* looks like a very active cell, despite the apparent lack of food[3]. The largest molecular difference between feeding and aggregating *A. kona* appears to be the induction of novel accessions, less than half of which are detected at 0 h (Supplementary Data 7).

## Starvation response or not

AGM development in the lab begins when food is depleted, or the amoebae migrate to food-depleted regions of the culture[3]. In either situation, the cells are presumably beginning to starve. This has led to the widely held belief that starvation is one of, if not the key inducer of AGM development[3,25]. Most microbes respond to adverse conditions such as starvation by encystation and dormancy[26]. However, AGM taxa can also respond by aggregating and building a multicellular fruiting body, potentially increasing the opportunity for, and extent of, dispersal. Eukaryotes use a number of pathways to gain energy and cellular building blocks when starved. Foremost among these are autophagy, where cells digest their own proteins and/or organelles, and proteolysis of ubiquitin-tagged proteins via the proteasome[27].

Autophagy is an ancient eukaryotic mechanism for cell survival during starvation[28,29]. It is also a major source of energy in aggregating Ddi AX4, particularly macroautophagy, the breakdown of mitochondria in vacuole-like autophagosomes[23]. The universal components of autophagy are not well defined as it has only been studied among Discoba in trypanosomatids[30], which are parasites with notoriously divergent protein sequences. Many autophagy proteins also show very low sequence conservation between the best-studied model systems, yeast, and humans, and even some autophagy proteins with the same annotation in both species show no discernible homology (see below). Therefore, we first predicted a universal autophagosome based on

**Table 2 | Differential gene expression in *Acrasis kona***

| category | Gro vs Agg | | Agg vs Germ | | full proteome |
|---|---|---|---|---|---|
| | **Aggup** | **Aggdn** | **Germup** | **Germdn** | |
| **A. general** | | | | | |
| Ako annotated accs | 370 | 260 | 1188 | 1641 | 10881 |
| Ako novel (unannotated) accs | 100 (22.3%) | 193 (42.6%) | 461 (28.0%) | 371 (18.4%) | 4987 (31.4%) |
| **Ako total Accs** | **448** | **453** | **1649** | **2012** | **15868** |
| Ddi AX4 SDE accs* | **2595** (1859–2078) | **1898** (2069–2375) | – | – | **11440** |
| **B. orthology groups (OGs)** | | | | | |
| Ako accs belonging to OGs | **190** | **198** | **880** | **948** | **6817** |
| Ako OGs | (162 OGs) | (176 OGs) | (558 OGs) | (657 OGs) | (2728 OGs) |
| OGs shared with Ngr | (35 OGs) | (18 OGs) | (104 OGs) | (133 OGs) | (360 OGs) |
| OGs shared with Disc | (75 OGs) | (35 OGs) | (162 OGs) | (232 OGs) | (605 OGs) |
| OGs unique to Ako | (52 OGs) | (123 OGs) | (292 OGs) | (292 OGs) | (1763 OGs) |
| Ako single copy accs | **258** | **255** | **769** | **1064** | **9051** |
| **C. transmembrane (TM) or signal peptides (SP)** | | | | | |
| Ako accs with TM domains | 67 | 83 | 371 | 371 | 3202 |
| Ako accs with SPs | 42 | 57 | 215 | 17 | 1712 |
| Ako accs with TM and SP | 9 | 15 | 62 | 62 | 439 |
| Ako accs without TM or SP | 349 | 328 | 11255 | 1686 | 11393 |

(A) Numbers of substantially differentially expressed accessions (accs) are shown for the transition from asocial feeding to social development (growth to aggregation - Gro vs Agg: Aggup/Aggdn) and from development back to asocial growth (aggregation to germination - Agg vs Germ: Germup/Germdn) (Supplementary Data 6). (B) Orthology groups (OGs) and their presence/absence in *Naegleria gruberi* (Ngr) and Discoba as a whole (Dis) were determined using OrthoMCL (Table S5 and Supplementary Data 2)[89]. Transmembrane domains and signal peptides were predicted using Phobius (Supplementary Data 1)[95].

abbreviations: accs (accessions), w/ (with), Ako (Acrasis kona), Ddi (*Dictyostelium discoideum* AX4), Ngr (*Naegleria gruberi*), Disc (Discoba), Gro (growth), Agg (aggregation), Germ (germination), Aggup/Aggdn (SDE increased/decreased from 0 h to 5 h aggregation), Germup/Germdn (SDE increased/decreased from 5 h aggregation to germination) (Supplementary Data 6). *Numbers of Ddi AX4 SDE accs averaged across two replicates (r1 + r2) each for 0 h vs 5 h starvation[11]. SDE acc numbers calculated for sing qle replicates (0h_vs_5h_r1 and 0h_vs_5h _r2) are shown in parentheses (GFold v1.1.4)[15].

Total numbers for individual categories in sections A and B are shown in bold.

annotation in the four model organisms where it has been studied: *Homo sapiens*, *Saccharomyces cerevisiae* S288C, *Schizosaccharomyces pombe* and Ddi AX4. The Ddi AX4 homologs were then selected for use as BLASTp queries because Dictyostelia is an outgroup to both animals and fungi, and its autophagy genes are strongly conserved relative to human.

The autophagy protein set of both Ddi AX4 and *Acrasis kona* appears to be very similar to that of humans, suggesting that these proteins constitute the ancient core autophagosome (Fig. 6). Moreover, many component proteins show high sequence conservation across these distantly related taxa including many accessions with BLASTp hits of e-100 or better between *A. kona* and human. During growth, *A. kona* also shows similar expression levels to Ddi AX4 for most of these proteins (0 h, Supplementary Data 9). However, there is no discernible increase in expression, much less SDE, for nearly all of these components during aggregation in *A. kona* (Fig. 6 and Supplementary Data 9). Moreover, autophagy overlaps with vacuolar assembly and transport, so the few *A. kona* SDE Aggup autophagy-like accessions could reflect other functions. In contrast, nearly all autophagy components are two- to four-fold Aggup in aggregating Ddi AX4 (Fig. 6 and Supplementary Data 9). Thus, it appears that aggregating Ddi AX4 makes extensive use of this system (Fig. 6)[23], while aggregating *A. kona* does not.

Another potential source of amino acid building blocks in starving cells is proteolysis, and the major cellular machinery for this is the proteasome[27,31,32]. Again, *Acrasis kona* encodes a complete and highly sequence-conserved proteasome (Fig. 7 and Supplementary Data 10). Most of the genes encoding proteasomal components are also moderately to highly expressed in both growing and aggregating *A. kona* cells, especially the proteasomal regulatory subunit Rpn11 (AKO1_008009, 1813 RPKM). However, only four of the 40 proteasomal protein genes are Aggup in *A. kona*, while two are Aggdn (Fig. 7). In contrast, although Ddi AX4 shows moderate to moderately high expression of proteasomal protein genes during growth, nearly every one of these genes shows at least a 2-fold increase in expression during aggregation (Fig. 7). Thus, Ddi AX4 appears to markedly increase protein degradation during aggregation, while the acrasid does not. Proteasomal protein synthesis in *A. kona* does nonetheless appear to be strongly regulated as the synthesis of the entire set of proteasomal protein genes appears essentially shut down in germinating spores (Fig. 7).

One degradative system that appears to be very active in all three life-cycle stages of *Acrasis kona* is the exosomal RNA degradation machinery (Supplementary Fig. S5 and Supplementary Data 11)[33]. This includes Dis3L/RPR44 (AKO1_011571), the RNAase component of the cytosolic exosome, for which there are over 4800 RPKM in all three life cycle stages (Supplementary Data 5). The cytosolic exosome plays central roles in translation regulation, mRNA quality control, transposon suppression, viral defense, and gene regulation by RNA interference (RNAi)[34]. *A. kona* appears to encode nearly all universal components of RNAi[35], some of which appear to be constitutive if moderately, expressed (Supplementary Data 11). However, the level of exosome gene expression appears very similar between the two amoebae, and neither shows signs of a marked up-regulation of the system during aggregation. Thus, we see no evidence in either amoeba of increased exosome production, which could potentially help augment the cell's nucleotide pool during starvation.

Forty-three *Acrasis kona* Aggup accessions are annotated as cell-cycle related, including accessions annotated for roles involved in regulation (4 accessions), cell division (8 accessions), DNA replication/repair (11 accessions) and nucleotide biosynthesis (10 accessions) (Supplementary Data 12). This suggests a possible role for cell division in *A. kona* development. Cell division could be useful for taxa that struggle to find sufficient aggregation partners to build a viable sorocarp[36]. Therefore, we reconstructed the *A. kona* kinetochore and examined its expression (Supplementary Fig. S6 and Supplementary Data 13). *A. kona* encodes a largely sequence-conserved kinetochore[37],

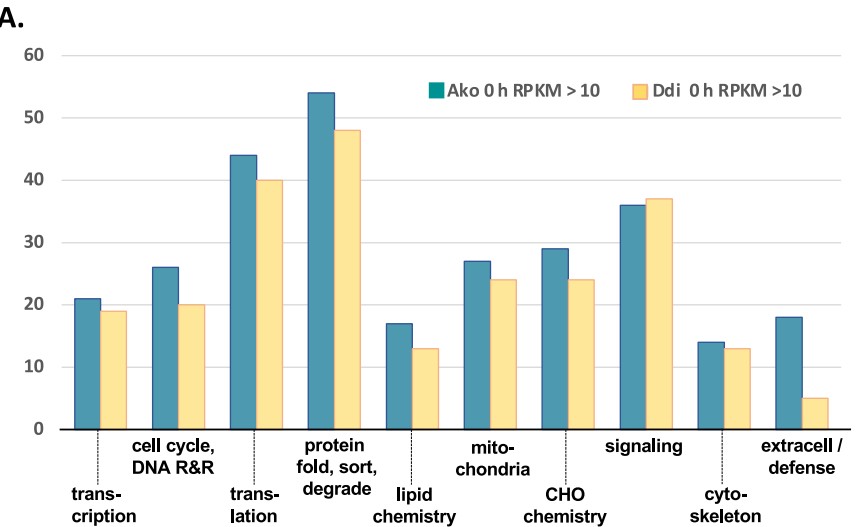

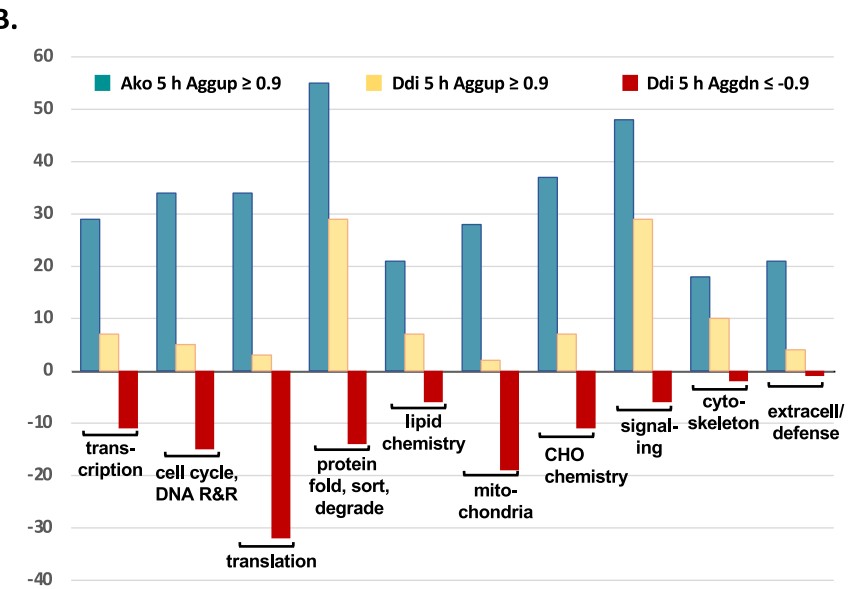

**Fig. 5 | Function profile of *Acrasis kona* developmental (Aggup) genes and their homologs in *Dictyostelium discoideum AX4* (Ddi AX4).** Numbers of predicted proteins distributed among 10 general function categories are shown for *A. kona* Aggup accessions and their Ddi AX4 homologs. Numbers are represented by bar height for (**A**) *A. kona* Aggup accessions and their Ddi AX4 homologs expressed during active growth (0 h, RPKM > 10) and (**B**) *A. kona* Aggup accessions and their Ddi AX4 homologs after 5 h of development (mid-aggregation). Evidence supporting the functional classification of proteins and their expression levels in *A. kona* and Ddi AX4 are given in Supplementary Data 7. Category abbreviations are as follows: DNA replication and repair (DNA R&R), protein modification, folding, sorting and breakdown (protein fold, sort, degrade), carbohydrate chemistry (CHO chemistry), extracellular or secreted (extracell/secrete). Differential expression of Ddi AX4 genes was calculated using data from ref. 11 and employing the same criteria as for *A. kona* (Supplementary Data 6).

although it appears that *A. kona* is unique among examined eukaryotes in lacking the key regulatory protein Cdh1 (Supplementary Fig. S7). Instead, the acrasid has two copies of the closely related Cdc20 (Supplementary Fig. S7), suggesting that the two proteins may functionally substitute for each other. However, there is little indication that aggregating *A. kona* is actively dividing. While a few kinetochore accessions show some increased expression during aggregation, a similar number show a decrease (6 Aggup, 7 Aggdn; Supplementary Fig. S6), and nearly all kinetochore accessions are expressed at only low to moderate levels (1.6 – 44.0 RPKM) (Supplementary Data 13). There is also no report of the *A. kona* sorocarp containing more cells than the initial aggregate.

Nonetheless, *Acrasis kona* Aggup accessions include DNA polymerase subunits α and δ, five replication complex proteins (ginsA, DNA topoII, RNAse HII, mcm3, cdk45), and 13 cell-cycle related accessions including five regulatory kinases (cdk2, cdk5, cyc_U4, Aurora, nimA). Many of these accessions also show moderate to high levels of expression during aggregation (Supplementary Data 13). Thus, although DNA replication is probably not leading to mitosis in aggregating *A. kona*, reaching an advanced cell-cycle stage may nonetheless be important. Cell-cycle stage influences cell fate in Ddi AX4 development, with cell-cycle advanced cells more likely to end up as viable spores rather than dead stalk cells. This is thought to indicate that late cell-cycle stage cells make more robust spores[38]. The large component of DNA replication and cell cycle annotations among *A. kona* Aggup accessions suggest that cell cycle stage may also play a role in assembly of the acrasid sorocarp. Although all aggregating *A. kona* amoebae form viable resting stages (aerial spores or stalk cysts[3]), the two cell types may have quite different fates. For example, aerial spores are more likely to be dispersed and to do so individually, while

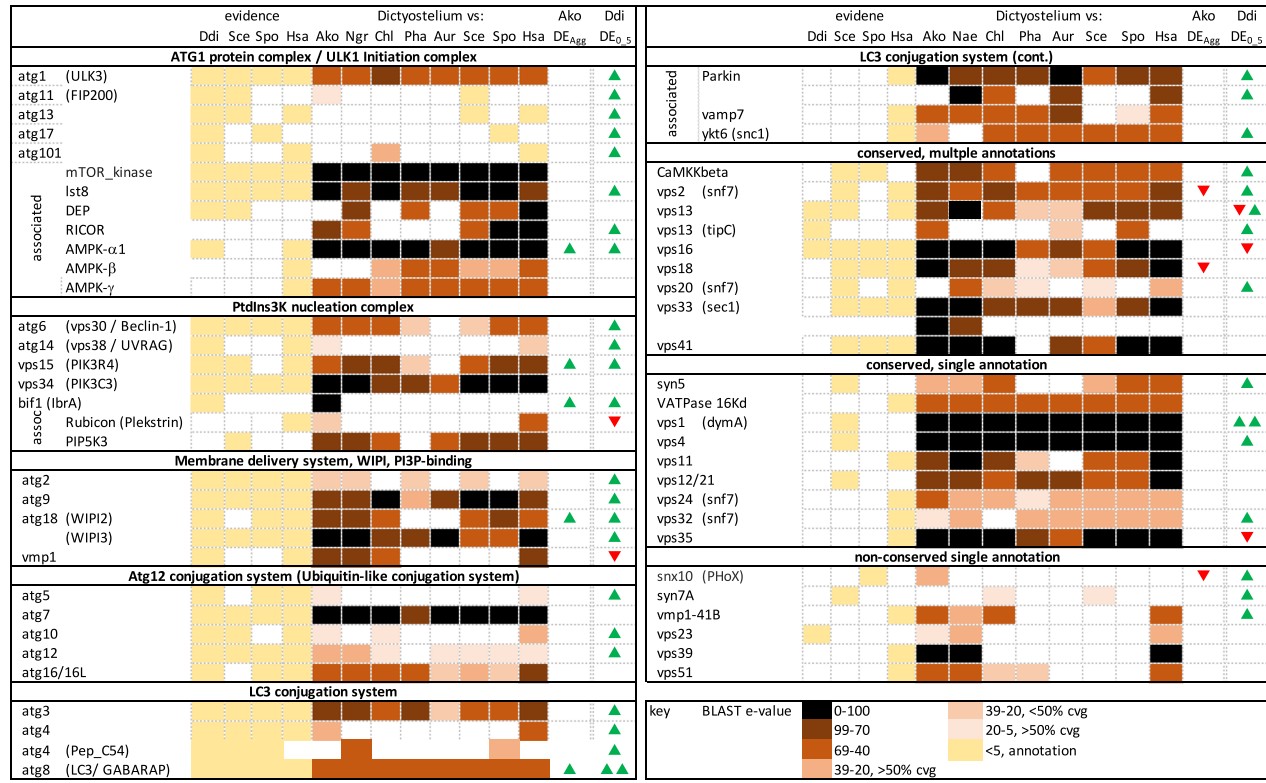

**Fig. 6 | The eukaryotic autophagosome and its expression in aggregating *Acrasis kona* and *Dictyostelium discoideum* AX4 (Ddi AX4).** A potential core set of eukaryotic autophagosomal proteins was predicted using annotation for *Homo sapiens, Saccharomyces cerevisiae S288C, Schizosaccharomyces pombe,* and *Dictyostelium discoideum* AX4 (Ddi AX4). Since all of the predicted proteins were present and well-conserved in Ddi AX4, these sequences were used as queries to identify homologs in other eukaryotes by BLASTp. Levels of sequence similarity are indicated by color intensity (key at the bottom right); weaker BLASTp hits ($e$-value $> e^{-39}$) were scored as present only if their BLASTp alignments included $> 50\%$ query coverage (cvg). Substantially increased or decreased gene expression is indicated by green and red arrows, respectively, for aggregating *A. kona* (Ako $DE_{Agg}$) and Ddi AX4 (Ddi $DE_{0.5}$). Taxon names are abbreviated as follows: Ako (*A. kona*), Aur (*Aureococcus* spp.), Chl (*Chlorella* spp.), Ddi (Ddi AX4), Hsa (*H. sapiens*), Ngr (*Naegleria* spp.), Pha (*Phaeodactylum* spp.), Sce (*S. cerevisiae*), Spo (*S. pombe*). Full details in Supplementary Data 9.

stalk cysts are more likely to remain behind and to do so as a group. This could lead to a selection process related to the suitability of different cell-stage cells for different fates.

## Developmental signaling

Cell signaling, both within and between cells, is critical for development, and a broad sampling of *Acrasis kona*'s extensive repertoire of signaling domains (Fig. 4) is Aggup (Fig. 8). For external signaling, the *A. kona* Aggup repertoire includes a single G-protein coupled receptor (GPCR, Supplementary Fig. S8) and two G-protein beta subunits (Gβ), one complete and one partial hybrid histidine kinase receptor (hybrid hisKR), two blue light receptor proteins (BLUF), two plasma membrane (PM) calcium pumps (Ca-ATPase_IIb), two lipocalin-interacting receptors (LMMBR) and three predicted single transmembrane domain (1TM) receptors (Fig. 8). Five of these receptors - the GPCR, both hisKRs and two of the 1TM receptors - appear to be largely aggregation specific (Germdn − 0.98 to − 2.81, Supplementary Data 14). The *A. kona* genome as a whole is particularly rich in histidine kinase signal receptors (Fig. 4), although the two Aggup hisKRs are unusual in showing the strongest similarity to bacterial sequences (*e*-value 0.0, Supplementary Data 14). The presence of BLUF protein sequences among the *A. kona* Aggup accessions are consistent with the fact that blue light is an inducer of *A. kona* development[39].

Internally, aggregating *Acrasis kona* upregulates major components of the cyclic AMP (cAMP), phosphoinositide (P_int), sphingosine (S1P), mitogen-activated protein kinase (MAPK), target of rapamycin (TOR) and assorted small GTPase pathways, albeit with some unique variations (Fig. 8 and Supplementary Table S7). *A. kona* cAMP signaling

most likely uses an essentially constitutive membrane-bound and sequence-divergent adenylate cyclase (AKO1_008377) along with Aggup versions of the main cAMP interacting partners CAP (adenylate cyclase associated protein), PKA (cAMP-dependent kinase) and two cAMP degrading phosphatases (PDE). For the P_int pathway, *A. kona* upregulates two versions of the key enzyme phospholipase C (PLC), which generates the secondary messengers inositol 3-phosphate (IP₃) and diacylglycerol (DAG). One of the main functions of IP₃ is binding to the IP₃ receptor in the endoplasmic reticulum (ER) membrane. This results in a flush of calcium release from the ER, the main site of calcium storage in eukaryotic cells. Although there is no apparent ER IP₃ receptor among the *A. kona* Aggup proteins, nor was any detected by BLASTp search of the full *A. kona* proteome, the protein is also not particularly sequence conserved. Moreover, there are at least four Aggup accessions predicted to encode novel multi-TM proteins, one or more of which could potentially fill this role (Supplementary Data 7).

These internal signaling pathways also appear to be tightly regulated in *Acrasis kona* as their Aggup components include both synthetase/activators and degradase/de-activators, particularly for the cAMP, S1P, Ras, and TOR (Arf) pathways (Fig. 8 and Supplementary Table S7). Several of these pathways converge on similar targets. For example, both the cAMP and TOR pathways likely affect cytoskeletal organization, suggesting changes in cell motility during *A. kona* aggregation. This is despite the fact that no marked outward changes in motility are observed in aggregating *A. kona*, such as elongation in the direction of aggregation as seen in Ddi AX4. In another example, both the RAS and S1P pathways converge on the MAPK signaling

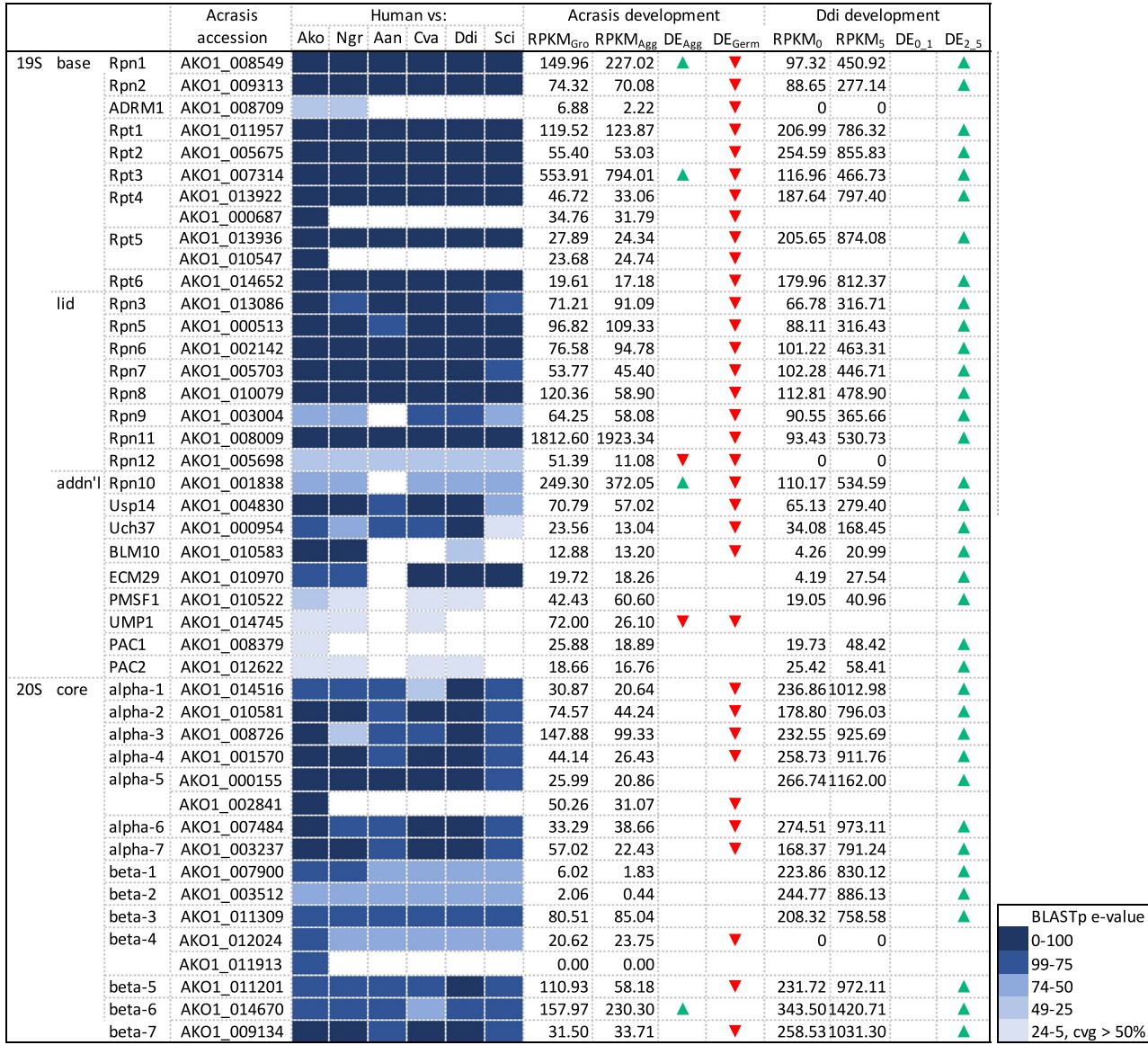

**Fig. 7 | The *Acrasis kona* proteasome and its life-cycle dependent expression pattern compared to *Dictyostelium discoideum AX4* (Ddi AX4).** Proteins were identified by BLASTp using human queries with hit strength indicated by color intensity (key at bottom right). Hits with low *e*-values ($e^{-25}$ to $e^{-5}$) were scored as present only if they showed > 50% query coverage (cvg). RNAseq length-corrected read numbers are shown for growing ($RPKM_{Gro}$) and aggregating ($RPKM_{Agg}$) *A. kona* cells and similar time points for Ddi AX4 ($RPKM_{0h}$ and $RPKM_{5h}$, respectively). Green and red arrows indicate substantially increased or decreased gene expression for aggregation overgrowth ($DE_{Agg}$) and germination over aggregation ($DE_{Germ}$) in *A. kona* and for 0h vs 1 h ($DE_{0\_1}$) and 2 h vs 5 h ($DE_{2\_5}$) hours of starvation in Ddi AX4. Accession numbers, BLASTp *e*-values, and DE values are in Supplementary Data 10. Taxon name abbreviations are as follows: *A. kona* (Ako), *Aureococcus anophagefferens* (Aan), *Chlorella variabilis* (Cva), *D. discoideum AX4* (Ddi), *Naegleria gruberi* (Ngr), *Saccharomyces cerevisiae S288c* (Sce). DdiAX4 gene expression values were derived from Santhanam et al.[11].

cascade, which includes two Aggup MAPK homologs. The most likely targets of these kinases are regulators of gene expression[40].

External aggregation signaling is expected to be largely species-specific as it involves attracting and screening for closely related cells. Thus, it is not surprising that the dictyostelid and putative acrasid external signal receptors appear to be more analogous than homologous (Fig. 8 and Supplementary Data 14). Internal developmental signaling, on the other hand, shows extensive homology between the two amoebae, including major components of the P_int, cAMP, SP1, RAS, TOR, and MAPK signaling pathways (Fig. 8 and Supplementary Data 14). Both amoebae also induce both internal and plasma membrane calcium pumps suggesting the possibility that both amoebae may excrete calcium as a chemoattractant (acrasin). Calcium is, in fact, a potent attractant for aggregating Ddi AX2, nearly as effective as their

better-known acrasin, cAMP, albeit over shorter distances and time scales[41]. Calcium signaling alone is probably insufficient to assemble the relatively large aggregates characteristic of most species of the genus *Dictyostelium*. However, it could be sufficient for the smaller acrasid aggregates, which also tend to be found on physically limited substrates[2,3]. It is also interesting to note that, although all species of *Dictyostelium* can use cAMP as an acrasin, the bulk of dictyostelid diversity, most of which build much smaller sorocarps, do not aggregate in response to cAMP, and their true acrasins are largely unknown[36].

## The extracellular matrix
Extracellular matrices (ECM) are central to membrane-level interactions in both clonal and AGM development[42]. In acrasids and

**A**

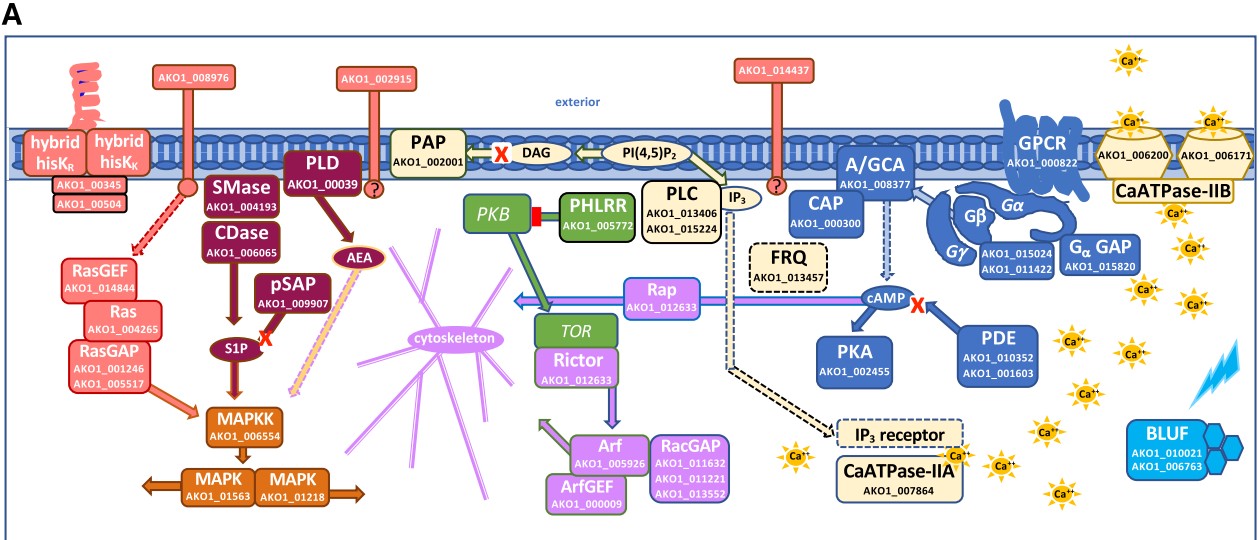

**B**

| signaling pathway | | protein | | *Acrasis kona* | | | Ddi AX4 |
| pathway | components | protein name | acronym | accession | RPKM Agg | DE Log2 Agg/Gro | DE Log2 0_5 |
|---|---|---|---|---|---|---|---|
| calcium | external | calcium ATPase-IIB PM | CaATPase-IIB | AKO1_006171 | 103.02 | 1.16 | ▲ 1.78 |
| | | calcium ATPase-IIB PM | CaATPase-IIB | AKO1_006200 | 299.10 | 1.00 | ▲ 1.78 |
| | internal | calcium ATPase-IIA2 Golgi | CaATPase-IIA2 | AKO1_007864 | 130.15 | 1.30 | ▲ 1.12 |
| | | laminin G domain Ca_binding | | AKO1_009084 | 120.27 | 1.25 | - |
| | | frequenin-1 | FRQ | AKO1_013457 | 22.85 | 1.54 | ▲ 1.63 |
| | P*inositide | P_int phospholipase C | PLC | AKO1_013406 | 31.43 | 2.79 | ▲ 1.95 |
| | | P_int phospholipase C | PLC | AKO1_015224 | 21.12 | 1.81 | - |
| | | Plekstrin domain + 11 LRRs | PHLRR | AKO1_005772 | 175.59 | 1.70 | ▲ 1.35 |
| | | phophoesterase | PAP | AKO1_002001 | 16.83 | 1.58 | ▼ -1.57 |
| GPCR | receptor | novel GPCR | GPCR | AKO1_000822 | 177.85 | 1.08 | - |
| | G proteins | G_protein alpha subunit | Gα | AKO1_008432 | 545.81 | *0.74* | ▲ 4.44 |
| | | G_protein beta subunit | Gβ | AKO1_015024 | 207.26 | 1.04 | ▲ 1.46 |
| | | G_protein beta subunit | Gβ | AKO1_011422 | 2712.08 | 0.60 | ▼ -1.21 |
| | | G_protein_GAP | Gα_GAP | AKO1_015820 | 14.18 | 2.32 | - |
| | cAMP/GMP | AMP/GMP cyclase | AKA | *AKO1_008377* | *167.34* | *0.33* | ▼ -2.27 |
| | | adenylyl cyclase associated | CAP | AKO1_000300 | 176.66 | 0.98 | 0.74 |
| | | cAMP protein kinase | PKA | AKO1_002455 | 66.92 | 1.15 | ▲ 1.37 |
| | | cAMP phophodiesterase | PDE | AKO1_001603 | 147.74 | 0.96 | ▲ 3.31 |
| | | possible PDE | PDE | AKO1_010352 | 33.40 | 2.13 | 0.43 |
| | | Rap, Ras related GTPase | Rap | AKO1_012633 | 18.30 | 2.29 | ▲ 1.80 |
| blue light | | BLUF domain protein | BLUF | AKO1_010021 | 126.63 | 1.17 | - |
| | | BLUF domain protein | BLUF | AKO1_006763 | 19.88 | 1.71 | - |
| hybrid hisK | | Hybrid his kinase/receptor | hybrid hisK | AKO1_003450 | 14.00 | 2.28 | ▲ 1.79 |
| | | Hybrid his kinase/receptor | hybrid hisK | AKO1_005045 | 10.51 | 2.29 | ▲ 1.79 |
| predicted | unk receptors | lipocalin receptor-like | | AKO1_012835 | 67.47 | 0.92 | ▲ 1.36 |
| | | lipocalin receptor-like | | AKO1_005686 | 540.5 | 1.41 | ▲ 1.36 |
| | | 1 TM, possible receptor | 1 TM | AKO1_008976 | 69.57 | 1.14 | - |
| | | 1 TM, possible receptor | 1 TM | AKO1_002915 | 42.63 | 2.36 | - |
| | | 1 TM, possible receptor | 1 TM | AKO1_014437 | 19.14 | 2.32 | - |

dictyostelids, the sorogen and sorocarp are enclosed in an ECM referred to as the slime sheath. The dictyostelid ECM appears to be something of a cellular dumping ground consisting of a wide assortment of proteins and protein fragments in a carbohydrate matrix. These proteins include enzymes with obvious ECM functions, such as carbohydrate metabolism and assorted proteases. However, in addition, there is a veritable alphabet soup of ~350 proteins or protein

fragments with no obvious extracellular function, e.g., numerous ribosomal proteins[43]. This has led to the suggestion that most of these proteins serve mainly to bulk out the ECM[43,44].

*Acrasis kona* Aggup accessions include 62 predicted proteins potentially involved in ECM construction (Supplementary Data 15). These proteins include nearly all major components of the major protein export machinery: coatomer-coated protein (COP) vesicles,

**Fig. 8 | Developmental signaling pathways up-regulated in aggregating *Acrasis kona*.** *A. kona* accessions with substantially increased expression during aggregation (Aggup) and several key components with substantial (RPKM > 100) but not substantially-increased aggregation expression are shown in (**A**) graphical and (**B**) table format. The table is further split into external (**B**) and internal (Supplementary Table S7) signaling pathways. Predicted proteins are grouped into common signaling pathways (www.expasy.org) with corresponding colors in (**A**, **B**) and Supplementary Table S7. Signaling interactions are indicated as follows: canonical activation (solid arrows), predicted activation (dotted-line arrows), inhibition (red-blockhead arrows), and degradation (arrows capped with a red X). Second messenger molecules are shown in ovals as follows: cAMP (cyclic AMP), DAG (diacylglycerol), IP 3 (inositol 3-phosphate), Pi(4,5)P 2 (phosphoinositol 4,5-bisphosphate), AEA (anandamide), S1P (sphingosine 1-phosphate) and calcium (Ca + +). Protein names are abbreviated as follows: G-protein coupled receptor (GPCR), G-protein alpha (Ga), beta (Gb), and gamma (Gg) subunits, hybrid histidine kinase signal receptors (hybrid hisK), mitogen-activated protein kinase (MAPK), N-acyl-phosphatidylethanolamine (PLD), small GTPases (Arf, Rac, Rap, Ras and Rho), GTPase exchange factors (GEFs) and activators (GAPs), phosphatase (P*tase), tyrosine kinase (Yk'ase). Structure/function predictions for AKO1_000820 (GPCR) and AKO1_011986 (novel 1TM receptor) are shown in Supplementary Fig. S8. Tables in (**B**) are organized by general signaling pathways and show, from left to right, *A. kona* accession numbers, protein names, top BLASTp hit e-values, DE $_{Log2}$ for aggregation overgrowth (Agg) and germination over aggregation (Germ), length corrected read numbers for aggregation (RPKM) and up-or down-regulation of Ddi AX4 homologs (green and red arrows, respectively). Details of sequence annotation, supporting evidence, and gene expression levels are shown in Supplementary Data 7 and 14.

clathrin-coated (CVV) vesicles, and vacuolar protein sorting/secretion systems. In fact, *A. kona* Aggup accessions include multiple copies of several components of each transport system (Supplementary Data 15). In addition, there are six secreted peptidases predicted along with two extracellular matrix stabilizing proteins and at least six likely-secreted Aggup proteins with ankyrin or TPR repeats that could serve as protease targets. Finally, there are two strongly expressed novel Aggup accessions with highly conserved vWFA domains, which are found in animal extracellular matrices and involved in cell adhesion (Supplementary Data 15)[45]. The *A. kona* Aggup peptidases include a highly-expressed aggregation-specific C26 peptidase, a predicted outer PM anchored zinc peptidase, an M60 peptidase, a tagC homolog, and two strongly expressed potentially secreted M49 di-peptidases (Supplementary Data 15).

Six of the 62 potential *A. kona* ECM proteins are highly expressed (RPKM > 1000), and another 18 are expressed with RPKM > 100 (Supplementary Data 15). This suggests the possibility that a substantial portion of the *A. kona* protein production during aggregation is devoted to constructing the acrasid's ECM. TagC is also interesting because of its critical role in the dictyostelid ECM, where it cleaves the pre-peptide SDF to release two small peptide hormones that control cell fate[46]. Overall, then, the acrasid SDE Aggup accessions could include a full suite of proteins to construct and maintain an ECM in a manner similar to that of dictyostelids and other multicellular eukaryotes[44].

**Novel developmental genes.** Altogether, there are 100 *A. kona* Aggup accessions with no known homologs in the GenBank nr database (Supplementary Data 7). Such accessions are good candidates for development-specific functions not required in non-developing relatives, such as ECM construction, spore coat synthesis, and transcription factors needed to promote and coordinate developmental gene expression. They are also candidates for species-specific functions that could be co-opted for development, such as quorum sensing and attracting and recognizing kin. Of particular interest are the 36 novel Aggup accessions that are largely aggregation specific (≥ 2-fold Aggup and Germdn, Supplementary Data 16), referred to here as Agg-specs. Four of these novel accessions are predicted membrane receptors, which are good candidates for species-specific sensing and signaling or cell-cell interaction. Another three novel Agg-specs are predicted to encode external membrane-anchored proteins, which are candidates for cell recognition markers, ECM anchors, or adhesion proteins. Three novel Agg-specs also carry signal peptide sequences, which tend to occur on secreted proteins such as would be involved in ECM construction, cell adhesion and/or spore coat synthesis. Eight of the novel Aggup accessions are strongly expressed (100–1000 RPKM), indicating proteins likely to be required in abundance, such as cell-surface markers or structural components of an ECM or spore coat. At the other end of the spectrum are three novel Agg-specs that are expressed at very low levels (5–10 RPKM) as expected for transcription

factors. Altogether, this small number of novel Agg-specs could potentially account for much of the specializations required for AGM.

Gene duplication is also a form of novelty and has been shown to facilitate developmental stage-specific gene expression and functional specialization in plants and animals[47,48]. *Acrasis kona* is rich in multigene families, many of which appear to be specific to acrasids, as they are either single copy or absent from a small but taxonomically diverse set of Discoba (Supplementary Table S5). Although *A. kona* Aggup accessions are not enriched for multicopy genes relative to the other life cycle stages examined here, 190 Aggup accessions belong to multigene families (Supplementary Table 2B). For nearly three quarters of these accessions, they are also the only Aggup member of their respective OG, that is, of the 190 Aggup OG accessions, 137 are the only Aggup family member (Supplementary Data 17). However, there are also 25 *A. kona* orthology groups with multiple Aggup members, and 17 of these OGs consist of just two members, so the full OG is Aggup (Supplementary Data 17). Thus, a fairly small set of novel genes and gene family expansions may constitute much of the molecular invention that accompanied the evolution of development in acrasids.

## Discussion

We have sequenced the genome of the aggregative multicellular amoeba, *Acrasis kona*, along with transcriptomes from the three main life cycle stages: growth, aggregation and germination (Fig. 1). The ~ 44 Mb genome is predicted to encode 15868 proteins (Supplementary Data 18), ~ 1/3 of which are novel and nearly half of which cluster into multi-protein families (Table 2B and Supplementary Data 2). The amoeba has broad metabolic (Supplementary Fig. S1) and signaling (Fig. 4) repertoires, as well as a large and functionally diverse set of phylogenetically-supported HGT accessions of diverse origins (Fig. 2 and Supplementary Fig. S4). In the switch from active feeding to aggregation and development, *A. kona* shows substantially increased expression of less than 3% of its protein-coding capacity (Table 2A) and little change in its metabolic activity (Fig. 5). This is in sharp contrast to the AGM model Ddi AX4, where aggregation affects nearly a third of the genome, major metabolic systems are shut down (Fig. 5), and the cells appear to be starving (Figs. 6 and 7)[17]. Nonetheless, the two amoebae show very similar signal transduction profiles during aggregation (Fig. 8 and Supplementary Table S7) and may use similar enzymes and pathways to construct an extracellular matrix (Supplementary Data 15).

### Acrasis kona development

The 448 *A. kona* Aggup accessions can be roughly grouped into housekeeping genes, potential developmental genes, and novel genes. The developmental genes include three major aspects of development, which are also the main areas of homology between the Aggup genes of *A. kona* and Ddi AX4: cell signaling, motility, and extracellular matrix construction (Fig. 5B). In fact, homologs of many of these accessions also play central roles in metazoan development[49]. The

strong conservation of these sequences from acrasids to dictyostelids and human indicates that they are very old and probably play essential roles in many, if not most, eukaryotes. However, the bulk of eukaryotes are probably strictly microbial. Thus it should not be surprising that about half of these apparent developmental accessions are not aggregation-specific in the acrasid, i.e., their expression is not substantially decreased in germinating cells (Germdn > − 0.9), if at all (Supplementary Data 7). It appears, then, that the acrasid employs largely ancient universal eukaryotic pathways for its development, with stage-specific roles induced for a small set of what may be key aggregation-specific components (e.g., Fig. 8).

Nonetheless, the advent of at least a few novel genes was almost certainly critical for the evolution of acrasid development. This is especially likely for aspects of aggregation that are almost certainly aggregation- and/or species-specific, such as cell-cell recognition, quorum sensing, spore and hila construction, and transcription factors and/or regulators. Good candidates for these roles can be found among the 36 accessions in *A. kona* that are novel and aggregation-specific. These include external membrane-anchored proteins suited for roles in cell-cell recognition and quorum sensing, secreted proteins that could be involved in constructing external structures such as the hila and spore coat, and lowly expressed globular proteins as expected for transcription factors and regulators (Supplementary Data 16). Additional acrasid genomes should help clarify the evolutionary rates and possible roles of these genes. For example, evolutionary rates should be especially high for cell-cell recognition and, possibly, quorum sensing genes.

## Acrasis vs Dictyostelium

The developmental cycles of acrasids and dictyostelids are remarkably similar. Both amoebae are commonly induced to develop in the lab by starvation, at which point growth ceases and aggregation begins. During aggregation, the cells gather together to build a mound of ECM-encased cells (the sorogen), which either continues to migrate as a unit or develops in situ. Once stationary, the sorogen transforms into a sorocarp consisting of a stalk supporting an aerial spore mass. In acrasids and the majority of dictyostelids, the sorocarp stalk is cellular, and aerial and stalk cells are differentiated, although dictyostelid stalk cells are dead at maturity whereas acrasid stalk cells remain viable. Cell fate is also not random in dictyostelids, with more cell-cycle advanced cells being more likely to form spores[38].

Dictyostelid cell fate is intriguing as ~ 20% of the aggregating cells, all formerly free-living and independent, self-sacrifice to form the sorocarp stalk. Although both acrasid stalk and spore cells remain viable, their morphological differences suggest they may also have different fates and use some type of selection process to decide among them. Mature acrasid stalk cells (stalk cysts) are irregularly shaped cysts, while mature aerial cells (aerial spores) are smooth and rounded with a thick spore coat[3]. Also, unlike stalk cysts, which tend to be parenchymatous, aerial spores are tenuously joined to one another by 2-3 small hila, which appear to form break points facilitating disintegration of the mature aerial array. Thus, aerial spores may be more likely to be dispersed and to do so as individuals, while cysts most likely remain behind and as a unit. This should make stalk cysts more likely to survive and to do so with sufficient relatives to form a future sorocarp, while spores are likely dispersed randomly to found new colonies, most of which are probably unsuccessful. This leads to the possibility that the suitability of aggregating acrasid cells for either fate could be related to some differentiating factor, such as their cell cycle stage, as is thought to be the case for Ddi AX4[38]. This possibility is consistent with the relative abundance of Aggup cell-cycle accessions in the absence of cell division (Supplementary Data 12).

There are also major differences between acrasid and dictyostelid development. Acrasid aggregation involves less than a hundred cells that migrate individually to form the sorogen, while dictyostelid

aggregation is generally a highly coordinated affair with 100 s to 1000 s of cells gathering into distinct streams that gradually coalesce in predictable patterns[25,50]. Dictyostelid sorogens can then migrate considerable distances, while sorogen migration in acrasids is very limited when it even occurs[3]. Dictyostelid spores also adhere to each other and disperse as a unit so that a new locale is founded by a relatively-large colony of closely related cells rather than individual spores. The acrasid amoebae themselves are aggressive micro-predators, much larger (~ 8–10 fold) and faster than dictyostelids, and probably mostly prey on eukaryotic microbes (Supplementary Movie 1)[3], while dictyostelids are primarily bacteriovores[25].

We also find that gene expression during development is markedly different between the acrasid and dictyostelid. After five hours without food, Ddi AX4 has substantially increased or decreased expression of nearly a third of its coding capacity[51], compared to less than 6% for *Acrasis kona* (Aggup + Aggdn, Table 2A). Moreover, much of the Aggup response in *A. kona* concerns information processing and housekeeping proteins, nearly all of which are strongly downregulated in aggregating Ddi AX4 (Fig. 5B). Aggregating *A. kona* also does not substantially increase expression of accessions predicted to be involved in autophagy or ubiquitin-linked proteolysis, while aggregating Ddi AX4 does both and to a very large extent (Figs. 6 and 7). Due to the conservative nature of GFold[15], we are probably missing some *A. kona* Aggup accessions expressed at very low levels. However, the striking similarity in the number of genes we find Aggup in *A. kona* (448) and the number of genes upregulated during non-starving development in Ddi (~ 500[52]) suggests that we are probably close. In general, then, aggregating *A. kona* looks like a very active cell undergoing a major life cycle transition with large increases in protein and energy production while aggregating Ddi AX4 appears to be in survival mode. This suggests that much of the difference in magnitude between the two amoebae's aggregation response may be attributable simply to a complex starvation response in the dictyostelid rather than greater developmental complexity. For the dictyostelid this probably includes additional pathways besides autodigestion (Figs. 6 and 7), such as switching to more energy-efficient forms of information processing and other housekeeping functions. Thus, while dictyostelid development is at least moderately more physically complex than that of *A. kona*, much of this may come down to a fairly small number of genes.

## Aggregation signaling

Extracellular signaling is central to AGM. For *Dictyostelium discoideum*, and probably all other known dictyostelids, this includes attracting cells, assessing co-aggregates for relatedness, monitoring aggregation size, and organizing the cells into a sorocarp. An especially critical feature of AGM is attracting suitable co-aggregates, for which dictyostelids use small diffusible molecules (acrasins). The acrasin for species of the revised genus *Dictyostelium*[20] is cAMP, whose role in cell signaling was first discovered in *Dictyostelium discoideum*[53]. However, for most of the 150 + species of Dictyostelia, the acrasin appears to be glorin, folate, or, in most cases, an unknown or undetermined molecule[36]. However, calcium appears to be almost as powerful an acrasin as cAMP for *D. discoideum*, the only member of Dictyostelia in which it has been studied[41]. Thus the use of calcium as an acrasin could be considerably more widespread in Dictyostelia. All that is currently known of the *Acrasis kona* acrasin is that it is not cAMP[3]. One possibility suggested by the results presented here is that this role could be played by calcium (Fig. 8). *A. kona* Aggup accessions include two versions of the $IP_3$ synthesizer PLC, one of which is highly aggregation specific (AKO1_013406), and two strongly expressed plasma membrane calcium pumps (Fig. 8). Together, these could be sufficient for the release of a pulse of calcium ion from the cell into the extracellular space. Moreover, the Ddi AX4 homologs of these enzymes are also Aggup (Fig. 8). Calcium cannot be relayed and amplified as efficiently

as a degradable acrasin such as cAMP, which is alternately synthesized and degraded to create outwardly radiating waves of signal in the dictyostelid[54]. However, calcium signaling might be sufficient for gathering the relatively small numbers of cells needed to build an acrasid sorocarp, perhaps further assisted by the often physically limited substrates where acrasids are most commonly found[2,3].

*Dictyostelium discoideum* screens its co-aggregates for relatedness using antigen-like cell-surface TIGR proteins, with the result that only closely related cells cooperate to form a sorocarp[55]. This should be especially important in the dictyostelid, where roughly 20% of the cells are sacrificed to form the dead cellulosic stalk. Although no cells are sacrificed to build the acrasid sorocarp, it is likely that the cells still perform some type of screening of co-aggregates for relatedness or, at the very least, the right species. This might be especially important if it uses Ca++ to attract aggregation partners since the strong sequence conservation of its component proteins suggests that this is an ancient pathway and possibly universal among eukaryotes (Fig. 8). *A. kona* Aggup accessions include three novel proteins predicted to be anchored to the external cell surface by a single transmembrane domain (AKO1_009152, AKO1_000315, AKO1_010358), and all three are aggregation specific (Aggup 1.33–2.39. Germdn − 1.24 to − 2.03, Supplementary Data 16). AKO1_009152 is also moderately highly expressed at 122 RPKM. Although the latter still seems to be a fairly low level for a protein destined to be distributed across the surface of a large cell, cell-cell recognition is probably most important early in aggregation so that the responsible gene may be more highly expressed earlier in aggregation.

For monitoring aggregation size, *Dictyostelium discoideum* uses two different systems. During early development, amoebae use the presence in the medium of a glycoprotein secreted by starving cells (conditioned medium factor) to assess the density of nearby aggregation-ready cells[56]. Later in development, the dicyostelid uses a 450 kD counting factor complex to assess aggregate size, leading to the splitting of overly large aggregates to form separate fruiting bodies[56]. *Acrasis kona* probably also assesses cell density, both early and late in aggregation. Before aggregation begins, acrasids can simply encyst individually if cell density is low[3]. Once aggregation is underway, the acrasid may split an overly large aggregate into fragments that migrate apart to form separate sorocarps, with the result that *A. kona* sorocarps tend to be similar in size even in dense cultures[3]. The most obvious candidates for possible quorum sensors in *A. kona* is a hybrid hisK receptor/kinase and a second separate hisK kinase, both of which are over 4-fold Aggup and 2-fold Germdn (Fig. 8)[57]. Other possible candidates are three novel accessions (AKO1_011986, AKO1_008976, AKO1_014437), all of which are predicted to encode single transmembrane domain receptors and have a largely aggregation-specific expression (Aggup 1.14–2.39, Germdn − 1.19–2.81 and Supplementary Data D16). AKO1_011986, in particular, is moderately strongly expressed (331 RPKM) and predicted to form large internal and external membrane tails (Supplementary Fig. S8).

All this leads to the possibility that quorum sensing could be the main inducer of development in *Acrasis kona*, as the aggregating cells in the lab, at least, are probably not starving. *A. kona* commonly aggregates in the presence of food by cells migrating individually to the edges of the culture. This likely has the dual effect of bringing cells together and doing so under more controlled conditions than in the midst of actively feeding cells and live prey. This suggests that the induction of aggregative multicellularity, at least in the case of *A. kona*, may be largely a matter of having sufficient cells to form a viable sorocarp. If this is the case, then some type of quorum sensing could be central to the induction of acrasid development. There is no obvious reason why this could not also occur in dictyostelids. Although starvation is a strong inducer of AGM in the lab for all known species of Dictyostelia[23,50], whether it is the only inducer in the wild seems unlikely. Spore formation and dispersal are probably useful means of survival under a variety of adverse conditions, in which case having sufficient cells to form a sorocarp could be the prime limiting factor in both acrasids and dictyostelids.

## Internal developmental signaling

While extracellular aggregation signaling appears to be largely analogous between *Acrasis kona* and Ddi AX4, *A. kona* internal aggregation signaling appears to be largely homologous between the two amoebae. Like Ddi AX4, *A. kona* appears to use the P_int, MAPK, cAMP, Ras, TOR, and S1P signaling pathways during aggregation, and most of the major components of these pathways are substantially up-regulated or constitutively expressed in both amoebae (Fig. 8 and Supplementary Table S7). Many of these pathway components also show strong sequence conservation despite the large evolutionary distance between Discoba and Amorphea (Supplementary Data 14). Thus, *A. kona* Aggup sequences for at least one key enzyme of the cAMP, TOR, and S1P pathways match human and dictyostelid homologs with BLASTp e-values of e-100 or better, and both *A. kona* Aggup MAPKs hit both taxa with at least[49,58]. These pathways also play central roles in animal development[49,58]. Thus, the presence of these signaling pathways and their strong sequence conservation in acrasids suggests that these pathways evolved early in eukaryote evolution, after which they were repeatedly if sporadically, co-opted for roles in multicellular development.

The antiquity and strong sequence conservation of these signaling pathways indicate that these pathways are ancient and play essential roles in eukaryotic processes unrelated to development. In fact, in *Acrasis kona,* only a few key components of these pathways appear to be strongly aggregation-specific, while most of the components are expressed at the same or even higher levels during germination (Fig. 8 and Supplementary Table S7). Nonetheless, at least one component of each pathway does appear to be aggregation-specific in *A. kona* (≥ 2-fold Aggup and Germdn). This suggests that the acrasid is manipulating essentially constitutive signaling pathways during development by inducing a few key non-constitutive components. These could then be combined with novel membrane receptors, most of which show strong life-cycle stage-specific expression (Supplementary Data 16), to control aggregation, development, and possibly differentiation.

## Evolution of development in Acrasis kona

Aggregative and clonal multicellularity together have evolved over twenty times in eukaryotes[59], including at least once in nearly every major taxonomic division (Supplementary Table S8). The eight known instances of AGM are probably also an underestimate as sorocarps are mostly too small to detect in the field, and many species may not readily aggregate in the lab. Most known instances of AGM also involve, at most, a small group of closely related species with closely related non-aggregating relatives. This suggests that AGM tends to be evolutionarily ephemeral. This makes Dictyostelia, with an estimated age of 300–600 million years[60], over 150 described species, and probably a large hidden diversity[61], by far the most evolutionarily successful AGM clade known. Whether this is because of or despite its developmental starvation response is an intriguing question or even whether starvation is the main inducer in the wild. In contrast, clonal multicellularity has given rise to at least five large diverse and ancient clades: animals, fungi, red algae, land plants, and brown algae[59], as well as far less evolutionarily successful experiments such as the various multicellular lineages of volvocalean algae[62].

Acrasids are separated from all known multicellular taxa, including dictyostelids, by 1-2 billion or more years of evolution[1]. Nonetheless, the results here suggest that there is extensive similarity at the molecular level between development in these taxa. In fact, both animals and fungi have close aggregating relatives, *Capsaspora owckzarzaki* in the case of animals and *Fonticula spp.* in the case of fungi[59].

The similarities in fundamental developmental processes among these taxa suggest that much of the basic tool kit for multicellularity evolved early in eukaryotes. This includes internal and external signaling pathways, ECM construction, and probably also differential expression of multigene families. Thus, the taxonomically-wide, if sporadic distribution of multicellularity across eukaryotes suggests that they have been experimenting with multicellularity for much of their history. In this case, multicellularity is either surprisingly rare, not particularly advantageous in many circumstances or there are many examples yet to be discovered.

Whatever key factors led to the evolution of acrasid multicellularity, the result is that the molecular biology of its development is not overlaid by a complex starvation response. Moreover, the number of genes involved appears to be small, even allowing for the conservative nature of GFold DE calculations[15], and a large fraction of these *A. kona* aggregation genes appear to be performing simple housekeeping functions. Thus, the evolution of multicellularity in *A. kona* may have required little more than a few unique inventions against a large background of genomic redundancy. This makes *Acrasis kona* a very simple model for the study of universal eukaryotic pathways and how these pathways have been co-opted for the evolution of multicellularity.

## Experimental procedures

### Cell culture and DNA extraction
*Acrasis kona* ATCC strain MYA-3509[63] was grown on *Saccharomyces cerevisiae* in liquid culture or on CM + (Corn Meal Plus) agar plates. For DNA extraction, spores from mature *A. kona* sorocarps were inoculated and grown in Spiegel's liquid medium[64] in 250 ml flasks shaken at room temperature on a rotary shaker (120 cycles/min). Acrasid cells were harvested in 50 ml corning tubes after 48 h at a cell density of approximately $1 \times 10^5$/ml, at which point the yeast cells had flocculated. Harvested cells were transferred to Petri dishes without food and left for at least 1 h to allow the amoebae to complete digestion of residual yeast material and to settle and attach to the bottom of the plate. Cells were then washed three times with 10 mM phosphate buffer to remove the pellets of flocculated yeast and harvested by centrifugation. DNA was extracted using the Blood & Cell Culture DNA Kit (Qiagen) as described in Fu et al.[65].

### Genome sequencing and assembly
Sequences were generated from *A. kona* total DNA using 454 GS Titanium (Roche) and Genome Analyzer (Illumina) platforms. Library generation and sequencing for both the 454/Roche (FLX + shotgun) and Illumina systems (500 bp Pair-End & 2 kb Mate-Pair) were carried out according to the manufacturers' protocols. All reads were treated for base correction and quality trimmed using the FASTX toolkit (v0.0.13) (hannonlab.cshl.edu/fastx_toolkit/) and Trimmomatic (v0.32)[66] with default settings (Supplementary Table S1).

The full *A. kona* genome was initially assembled from 454 FLX + shotgun reads and Illumina paired-end reads as a de novo hybrid assembly using MIRA (v3.9.9)[67]. Illumina mate-pair reads were then used for scaffolding using SSPACE (v2.0)[68]. The assembly was filtered to remove redundant scaffolds, where redundancy was defined as scaffolds < 5 kb with a greater than 80% identity to another scaffold > 5 kb (Supplementary Table S1a). The quality and completeness of the genome assembly were assessed using QUAST[69], and CEGMA (version 2.4)[12], and BUSCO (version 3.0)[13] (Supplementary Tables S3).

### RNA Extraction and transcriptome sequencing
Cells were grown in Spiegel's liquid medium[64], and total RNA was extracted using TRI Reagent LS (Sigma-Aldrich). Stranded mRNA libraries were constructed separately from total cultures as well as germinating spores, actively growing cells, and aggregating cells (Fig. 1) using Illumina's Truseq RNA sample prep kit and sequenced on

a HiSeq2000 Illumina platform ($2 \times 100$ bp, Pair-End). Adapter sequences and low-quality bases were removed using Trimmomatic v.0.32[66] with default settings, and de novo transcriptome assemblies were generated using Trinity (version 2014-07-17). The latter followed the protocol by ref. 70 with the --jaccard_clip option set to "yes" to minimize the fusion of transcripts in a relatively gene-compact genome.

For the three different life stages (Fig. 1), sequence quality was checked using FastQC (v0.11.8)[71], and the quality trimmed sequences were aligned to the genome assembly with STAR (version 2.5.2)[72]. Reads mapping to genes were counted using default settings in the featureCounts program in R[73]. Since we had only a single RNAseq replicate for each life cycle stage, fold change between growth and aggregation (gro v agg), and, aggregation and germination (agg v germ) was calculated using GFOLD (v1.1.4) (Supplementary Data 5 and 6[15]).

### Gene prediction
Before gene prediction, low-complexity and interspersed repeat sequences were identified and masked using RepeatRunner[74] and RepeatMasker (www.repeatmasker.org/), the latter including novel and non-novel repeats from *Naegleria gruberi*[14]. Genes were predicted using Augustus (version 2.7)[75] and SNAP (November 2013 release)[76] in the Maker pipeline[77] using parallel strands of evidence-based and ab initio gene prediction (Figure S9). Evidence-based gene prediction utilized both protein and transcriptome data to train the gene predictors. Protein evidence consisted of the UniProt/SwissProt database and the predicted proteome of *N. gruberi* (JGI release Naegr1). Transcriptome data consisted of pooled 454 and RNAseq data.

Ab initio gene prediction used the same input as above, together with the resulting "evidence-based" gene models and a set of 862 best gene models selected using the PASA package[78] based on four criteria: (i) completeness (presence of start and stop codons), (ii) physical distance from other genes ( > 500 base pairs from any other predicted gene model), (iii) non-redundant coding sequence (removing isoforms or close homologs), and (iv) availability of RNAseq data for the full length of the gene.

A final round of gene prediction was then run using all raw input plus the results of both the evidence-based and ab initio builds to train the gene predictors and then passed through the MAKER pipeline again. This final step was performed in order to check the congruence of the predicted gene models from the evidence-based and ab initio strategies. The final result was four tracks of gene prediction: evidence-based gene predictions, RNAseq-based PASA predictions, ab initio predictions, and combined ab initio and evidence-based gene model predictions. These were then curated manually to identify and resolve any conflicts (Supplementary Fig. S9).

### Manual curation
The four gene prediction tracks were uploaded into Web Apollo[79] using the National Bioinformatics Infrastructure Sweden (NBIS) resources (https://www.nbis.se/). Each predicted gene was then checked by eye for congruence among the prediction methods. In cases of incongruence, all lines of evidence were re-checked, and the most reliable was selected. A gene model was considered reliable if it had a start and stop codons, canonical intron splice sites (in most cases), > 500 nucleotides 5' and 3' flanking sequence (UTR), and full RNAseq coverage and/or at least one Pfam functional domain[80]. Predicted genes with non-canonical splice sites were compared with RNAseq evidence and assembly coverage to check for possible errors and were otherwise kept. The predicted genes were checked for duplication using Python scripts. All gene duplicates with 100% identity at the nucleotide level were inspected manually by aligning their parent contigs using EMBOSS stretcher[81] and by checking their read coverage in Tablet[82]. This was followed by a final check

to identify and combine overlapping contigs and remove duplicate contig fragments.

## Gene function annotation

Functional annotation of the fully curated gene models was carried out using the publicly available NBIS functional annotation pipeline (https://github.com/NBISweden/pipelines-nextflow). Each predicted gene model was searched at the deduced protein level using BLASTp against the UniProt/SwissProt reference database in order to retrieve gene and protein names and predicted protein functions. The predicted proteins were also passed through InterProScan (version 5.7–48)[83] in order to retrieve InterPro, Pfam[80], and Gene Ontology (GO)[84] data. These metadata were then parsed into the genome annotation files using Annotation Information Extractor (Annie)[85] for viewing in WebApollo[79] (Supplementary Fig. S9). Gene ontology annotations for the predicted genes were also obtained using Blast2GO[86] with default settings to confirm the consistency of the results. Protein translations of the curated gene models were also searched using BLASTp against the NCBI non-redundant protein database with an e-value cut-off of 1e-5. Metabolic pathways were annotated for both *Acrasis Kona* and *Naegleria gruberi* using KEGG Orthology (KO) numbers using BlastKOALA[87]. This was also done for *Naegleria gruberi*. The resulting KO numbers were used as input to iPath[88] to compare the two metabolic maps.

## Gene families

Gene families were defined by all-vs-all BLASTp and clustering using OrthoMCL[89]. Clustering included proteomes from a broad taxonomic sampling of eukaryotes (Supplementary Table S5). OrthoMCL was used with default settings with an e-value cut-off of 1e-10 for BLAST and an inflation value of 1.5 for MCL. The analysis was then repeated with ProteinOrtho[90] with default settings (Supplementary Data 2). The results were processed using Perl scripts to identify the number of genes in each family for each organism.

## Horizontal gene transfer

Complete NCBI databases were downloaded from RefSeq (April 2016) for Metazoa, Fungi, Viridiplantae, Bacteria, and Archaea, and non-redundant protein databases for Amoebozoa, Stramenopila, Rhizaria, Alveolata, Rhodophyta, and Excavata (Discoba + Metamonada). Transcriptomes for *Percolomonas cosmopolites* strains AE and WS (Heterolobosea, Discoba) were downloaded from iMicrobe[91]. Transcriptomes of *Andalucia godoyi* and *Seculomonas ecuadoriensis* (Jakobida, Discoba) were generated locally[92]. Local individual BLAST databases were created for each of the major groups above, and the *Acrasis Kona* predicted proteins were queried against each database with an e-value cut-off of 1e-10. The analyses were repeated separately for *Naegleria gruberi* (JGI release Naegr1)[14] and *Dictyostelium purpureum* (assembly v1.0)[93] for comparison.

*Acrasis kona* proteins with hits only to Bacteria, Archaea, or any single major eukaryotic group (~ kingdoms, Adl et al.[10]) other than Excavata were screened for potential signal peptides using the SignalP (version 4.0)[94] and Phobius webserver[95]. Functional categorization of these proteins was done using GoFeat[96]. All non-novel *A. kona* proteins with non-excavate top hits and total hits (*e*-value < 1e-35) to three or more major eukaryote groups were analyzed phylogenetically using a bioinformatic pipeline as follows. (i) Each protein was searched against the local NCBI database for each major group separately, and all the hits with e-value < 1e-35 were retrieved. (ii) Sequences were aligned using MUSCLE[97] in AliView[98], and alignments were trimmed using the heuristic (automated1) algorithm in trimAL[99]. (iii) Maximum likelihood trees were constructed using IQTree[100] with standard model selection (-m TEST) and the SH-aLRT and ultrafast bootstrap (1000 replicates) tests. iv) Evolutionary relationships in each resulting tree were identified using SICLE[101].

## Developmental transcriptome analysis

Accessions with differential expression (DE) during *Acrasis kona* development were annotated based on a consensus of top BLASTp hits to 1) all GenBank taxa (nr database), 2) human (SwissProt) associated annotation, and 3) *D. discoideum AX4* (RefSeq) and associated Dictybase annotation (http://dictybase.org/[21]). Accessions were manually clustered into categories based on their function predictions, and also automatically annotated and clustered using BlastKOALA or, for larger data sets, GhostKOALA, with an unspecified target taxon (https://www.expasy.org/). Koala annotations per functional categories were then tabulated after the automated removal of duplicates. Transmembrane protein structure and orientation were predicted using TMHMM 2.0[102] and potential G-protein coupled receptors by GPCRHMM[103].

The proteasome, exosome, and kinetochore were predicted for diverse eukaryotes by BLASTp at NCBI using genus- or species-level taxid limits and human RefSeq queries. Homologous sequences for *Acrasis kona* were first identified by local standalone BLASTp and then confirmed by BLASTp against the human RefSeq database. For the autophagosome, a potential core protein set was first predicted by assembling a consensus of all GenBank autophagy annotated accessions for human, *Saccharomyces cerevisiae S288C*, *Schizosaccharomyces pombe* and *Dictyostyelium discoideum AX4*. Sequences for diverse eukaryotes were then identified by BLASTp using taxid limits and *D. discoideum* AX4 queries. Mean read counts for the normalized reads for each replicate for *D. discoideum* AX4 developmental genes were retrieved from Santhanam et al.[11]. Differential expression was calculated for zero, two, and five hours after food was depleted (starvation) using log$^2$-fold change between time points.

Phylogenetic analyses of potential aggregation-specific horizontal gene transfers were conducted at the amino acid level using sequences identified by BLASTp against GenBank nr. Sequences were aligned using MUSCLE (v3.8.32)[97] as implemented in AliView[98] and trimmed using trimAl (automated1 parameter)[99]. Trees were constructed using RAxML with the LG+gamma substitution model and bootstrapping with automated stopping[104] on the CIPRES webserver[105].

## Reporting summary

Further information on research design is available in the Nature Portfolio Reporting Summary linked to this article.

## Data availability

The data generated in this study, the *Acrasis kona* whole genome shotgun project, has been deposited at DDBJ/ENA/GenBank under the accession number JAOPGA000000000 (GCA_026419775.2). The version described in this paper is version JAOPGA020000000. Transcriptome data for this paper are deposited in SRA files SRR22861965 SRX18820705, SRX19285604, SRX19285605, and SRX19285603. *Acrasis kona* strain MYA-3509 (formerly *Acrasis rosea*) is available from the American Type Culture Collection (mya-3509) or from MWB upon request.

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

## Acknowledgements
We thank Robert Insall for the *Acrasis* movie and Wei Miao for generously funding the initial genome sequencing. Thanks also to Marc P. Höppner, Jacques Dainat and the Uppsala Science for Life (SciLife) lab for technical support along with Caesar Al Jewari, and the Uppsala Multidisciplinary Center for Advanced Computational Science (UPPMAX), the Swedish Bioinformatics Infrastructure for Life Sciences (BILs) and the Cyberinfrastructure for Phylogenetic Research (CIPRES) for use of computational resources. We also wish to thank the reviewers for their careful reading of the manuscript and their useful comments and suggestions. This work was supported by grants from the Swedish Research Council to SLB (2017-04351 Vetenskapsrådet, https://www.vr.se/). S.S. was funded in part by a grant from the Department of Organismal Biology, Uppsala University. M.W.B. was funded by the United States National Science Foundation (NSF) Division of Environmental Biology (DEB) grant 2100888 (http://www.nsf.gov).

## Author contributions
Conceptualization: S.L.B. Methodology: S.S., C.J.F., and S.L.B. Formal Analysis: S.S., C.J.F., and S.L.B. Data Curation: S.S., C.J.F., and S.L.B. Resources: M.W.B. and S.L.B. Data interpretation: S.S., C.J.F., M.W.B., and S.L.B. Visualization: S.S., C.J.F., and S.L.B. Funding acquisition: S.L.B. Project administration: S.L.B. Writing: S.S., C.J.F., M.W.B., and S.L.B.

## Funding

## Competing interests
The authors declare no competing interests.
