## [Transparent Peer Review file · Nature Communications]

The *Acrasis kona* genome and developmental transcriptomes reveal deep origins of eukaryotic multicellular pathways

Corresponding Author: Professor Sandra Baldauf

Version 0:

Reviewer comments:

Reviewer #1

(Remarks to the Author)

Sheikh, Fu, et al. present the genome of *Acrasis kona*, an amoeba in the evolutionarily important and understudied Heterolobosean phylum of eukaryotes. Acrasids have a remarkable life cycle that includes aggregative multicellularity and formation of a sorocarp, currently the only organisms in this part of the eukaryotic tree known to manifest these behaviors. The authors conduct in-depth analysis of the *A. kona* genomic repertoire, combined with transcriptomic analysis of major stages during transition to the multicellular stage. Comparing the multicellular transition of *A. kona* to that of *Dictyostelium discoideum*, the classic model of aggregative multicellularity, the authors find numerous differences in gene expression patterns, while identifying potentially shared signaling motifs between these two divergent organisms.

The genomic resources presented here are important and will certainly spur the field in unraveling the many cell biological mysteries remaining in this understudied eukaryotic group. The choice of this particular species for in-depth genomic analysis is all the more appropriate due to the intense recent interest in the evolutionary origins of eukaryotic multicellularity, for which *A. kona* currently provides a taxonomically distinct example from existing model systems with sequenced genomes. This work will be of significant interest for researchers in comparative and evolutionary cell biology who are tracing the development of multicellularity. Furthermore, the addition of another well-annotated genome in the Heterolobosea will also enable further studies of the nature of the last eukaryotic common ancestor. The paper is well written and reflects the breadth of questions that will now be accessible with the publication of this genome.

Unfortunately, although the authors have deposited the genome scaffolds in NCBI, the gene models have not been deposited, making confirmation of the reported findings impossible. Moreover, the poor presentation among several of the primary figures obscures many of the major contributions of the work and—more importantly—further impedes verification of many of the authors' claims. Resolving these issues will require depositing the gene models in NCBI, re-analysis of existing data, significant changes to the presentation of figures and underlying data, as well as textual changes, as outlined below:

Major points

1. In order for other scientists to assess the work, confirm findings, and build on these analyses, the authors must make the gene models publicly available. Currently, the NCBI accession provided only includes genome scaffolds and the RNAseq data sets, not the gene models. The "Accession IDs" associated with predicted protein sequences included in the supplement do not link to anything we can find in NCBI.

2. Our biggest concern with the analyses as presented (and as reviewable without access to the gene models) is with the validity of the claim that relatively few genes are differentially expressed during *A. kona* aggregation. As far as we can tell, this conclusion is based on RNA-Seq with only one replicate per condition. While the authors take care to use the most appropriate statistical tool for this scenario (GFOLD), nevertheless the lack of replicates necessarily results in more conservative differential expression calls. In comparing the number of differentially expressed genes in *A. kona* to a dataset from *D. discoideum* with multiple replicates, one might expect to find more differentially expressed genes in *D. discoideum* simply due to the greater number of replicates providing more statistical power. The methods suggest that the authors did

not perform any comparable analysis on the Dicty data to the Acrasis analysis: specifically, how many genes are differentially expressed for Dicty when considering only one replicate from the Dicty data, passed through GFOLD as with *A. kona*? This apples-to-apples analysis of RNA-Seq data run through the same statistical pipeline is critical for comparing differential expression between these two groups. This will require downloading the Dicty raw data from GEO and assessing only one replicate at a time using GFOLD.

3. A related point: the fact that only 1 replicate was used per condition is extremely important for the reader to appreciate the data, and this should be stated more prominently by explicitly indicating how many replicates were performed for each condition (currently the statement in line 676-677 “as replicates were unavailable for each stage” is ambiguous as to if replicates were available for SOME stages but not all).

4. The authors should also explicitly acknowledge the alternative hypothesis to many of their claims that that aggregation in *A. kona* (including genes like autophagy genes) is regulated post-transcriptionally, in contrast to *D. discoideum*.

5. The autophagy section is based on the assumption that *Acrasis* and related species undergo autophagy using the same genes/pathways as Opisthokonts and Dictyostelia. This is a major assumption and should be stated as such up front.

6. The HGT analyses appear to be based only on comparisons to predicted proteins from gene models. Gene prediction in these species is notoriously difficult, and the *Naegleria gruberi* genome is known to have errors in gene prediction. The authors should use tBLASTn or similar to search for the putative HGTs on scaffolds of *Naegleria* and other Discobans before asserting that these are “confirmed HGTs”.

7. The graphical abstract demonstrates the *A. kona* cytoskeleton using cytoplasmic microtubules and a cytoplasmic centrosome (including centrioles), yet neither of these has been observed in *A. kona*, and in fact there is some evidence that this organism may lack cytoplasmic microtubules like the related *Naegleria gruberi* (Roos UP and Guhl B. (1996) Eur J Protistol 32: 171-189). We propose that the authors choose a more accurate way to demonstrate the cytoskeleton by illustrating actin instead, since it is known to be abundant in *A. kona* cytoplasm (Hellstén M and Roos UP. (1998). Fungal Genet Biol 24: 123-145). Alternatively if the authors insist on using microtubules, then they should use the one currently documented microtubule structure in *A. kona*, the intra-nuclear spindle (Roos and Guhl, vide supra).

8. The utility of the pie charts in Figure S1 is unclear; it seems a well laid out table would be a more useful way for the reader to make comparisons (rather than scrunching that into the key).

9. The data underlying Figure S2 are not available, and the figure itself is missing a lot of information, such as labels for different metabolic steps.

10. The data tables are difficult to understand. Specifically, we were not able to use Figure 4 to verify a number of claims made by the authors: f

- Figure 4B is extremely difficult to interpret. 3D bar graphs are very troublesome for the reader to interpret in general, but because the bars on this graph extend below zero on the y-axis, it is in this case illegible and impossible to understand. The authors should re-create this graph using grouped bar charts in a conventional 2D presentation, or consider using UpSet plots (see below).

- Many assertions are based on adding or dividing different columns in figure 4A, which is a tiresome exercise for the reader, and in many cases we cannot verify the numbers in the text. Please re-do this table to make your points clear, and follow a standard convention that if there is a row in the table for “all” or “total”, then it should be clear which other rows in the table will sum to the “all” row. Some specific issues:

- For Aggup/Aggdn/Germup/Germdn, the number of accessions in ‘novel,’ ‘Multicopy,’ ‘cHGT,’ and ‘TM +/- SignalP’ don’t add up to ‘all.’ While we acknowledge that these categories don’t seem to be mutually exclusive, in that case we would then expect the sum to add up to more than the ‘all’ row due to double-counting. Instead, sometimes the sum adds up to less than the ‘all’ row (e.g. for Aggup), suggesting some genes in the ‘all’ category are not broken down! This is confusing. We encourage the use of horizontal lines to separate logically distinct groupings of mutually exclusive categories, and ideally within each grouping the sum of accessions should add up to ‘all’ at the top (Though perhaps an exception to this rule could be made for simple binary categories like ‘has TM’ and ‘does not have TM’)

- Line 157 states that Germup and Germdn show change of 1771 and 2033 accessions, respectively. We are unable to reproduce these numbers using the data in the table.

- Lines 164-165 mention 1214 OGs involved in growth and references Figure 4A. But Figure 4A shows differences between Agg, Gro, Germ, not Gro explicitly, so this claim cannot be verified from the table (is Aggdn + Germup taken to be synonymous with growth? If so, we still cannot get to 1214 OGs)

- In general the claims related to Figure 4A and 4B would benefit from an UpSet plot (doi:10.1109/TVCG.2014.2346248, can be made using the UpSetR package in R), which will allow the author to break the gene categories down into whichever granularity they choose and allow explicit comparison between claims in the text and specific bars in the table.

- For example, Line 205-207 describes the overlap between two categories: “Aggup” and “AKo 0 h RPKM > 10” based on the fact that bar charts are shown for both of these. But the overlap between these categories is not shown in Figure 4B, so this claim cannot actually be verified based on this data!

11. Beyond Figure 4 there are a number of claims that cannot be verified from the information provided. More information should be provided or claims should be adjusted.

- Line 175 states that “none of the 35 *A. kona* Aggup cHGTs trace to Dictyostelia based on BLASTp or phylogeny.” First, we count 36 Aggup cHGTs in Table S11. But more importantly we are not sure how to evaluate this claim. For example, from

just a cursory inspection it seems that at least 2 genes in Table S11 with a cHGT value > 60% have a strong BLASTp hit in Dictyostelium (AK_01251, nfyA, top hit in Dicty has e-value 9e-21 to a gene annotated nfyA; AK_14998 clp1, top hit in Dicty has e-value 7e-108). Are these genes ruled out by phylogeny? If so then how do we evaluate the phylogenetic screening (it does not seem to be shown in Table S11)? If the authors want to make a claim based on Table S11, it should be more explicit in the table, or the claim needs to be broken down into verifiable pieces.

- Line 348: how are we to understand analogy and homology from this diagram? The external signaling pathways are scattered across all the categories – it is very difficult to assess this claim. Perhaps the authors could mark which genes they are defining as pertaining to external signals to help the reader follow this (extremely busy and confusing) figure?
- Line 381-382: “13% of the aggregating cell protein production is potentially involved in ECM construction” This claim does not follow from the data – protein production is not measured. The authors could instead comment on the diversity of expressed genes or something like that.
- Line 395: where do these numbers come from? We can’t figure out how to get them from the table shown.

Minor points:

- Can the authors make any comments on the possible ploidy of *A. kona*, perhaps based on the proportions of SNVs in their sequencing data?
- Confused by Agg vs Germ and Gro vs Agg nomenclature: which is denominator? Aggup and Aggdn were clearer, maybe just explicitly state that Agg vs Germ = Agg / Germ or something like that.
- Figure 3: How did the authors decide to focus on this particular list of signaling domains?
- Line 99: *N. fowleri* also have a well-characterized genome at this point (2 papers, fairly extensive)? The authors should amend this point to indicate “one of only two members of the phylum Heterolobosea with a well-characterized genome (the other being the closely related and pathogenic *N. fowleri*)”
- Line 128-129 - It is a stretch to say that the HGT data “confirms” that acrasids are omnivorous micro-predators that “probably” acquire HGTs from prey. The authors should walk these back along the lines of: “consistent with being omnivorous” and “may acquire HGTs from prey.”
- Line 169-170: This claim would be strengthened by some quantitative or statistical argument supporting the use of the term “enriched;” a chi-square or Fisher exact test for the association between genetic redundancy/novelty vs. agg/growth/germ would be useful.
- Line 239: The authors construct a “universal” autophagosome based only on species from Amorphea... We suggest choosing a different word that is more reflective of the (necessarily) limited taxonomic sampling underlying this categorization. What about “consensus”?
- Line 275-277: Why does this appear unlikely? There is no evidence for discarding this hypothesis after spending a whole paragraph discussing it.
- Line 305: Is there any evidence that stalk cysts can de-differentiate into trophozoites? This seems relevant to the speculation in the Discussion about the fate of stalk cysts after spore dispersal (lines 468-469)
- Line 449: strong claim based on this data, especially since there is not any functional evidence linking any of these genes with phenotypes. The authors should soften this language.
- Line 453: We understand that “starvation” here means that food has been depleted, but this seems to undercut the notion that the *A. kona* cells are not starving while the *D. discoideum* cells are. Perhaps a different word would help disambiguate the two senses of ‘starvation’?
- Line 589: “the acrasid appears to be...” is too strong in the absence of functional data for any of these genes. To my ear “appears to be” signals an evident conclusion, for example in the previous sentence where you use “appears to be” to describe an observation from the data. Weaken this to “the acrasid may be...”

Typos

- Line 162: Table 4A → Figure 4A
- Line 445: Table 11 → Table S11
- Figure 4 legend, last sentence: “Differentially expression” → “Differential expression”
- Title of Figure S8 should be related to kinetochore or cell cycle not autophagy
- Figure 5: “evidene” should be “evidence”

Reviewer #2

(Remarks to the Author)

The authors report the genome sequence of the protist *Acrasis Kona*, a member of the Excavata clade. They also present transcriptomic sequences from three different moments of the life cycle, which includes an aggregative multicellular stage with fruiting bodies. These include growth, aggregation, and spore germination. The authors analyze the genes significantly upregulated in those time points and compare their function to those found in *Dictyostelium*, which also develops into multicellular fruiting bodies. The authors suggest that *A. kona* uses very similar pathways on its development as *Dictyostelium*.

The community will be very pleased with the genome and transcriptomic data here reported. The analyses appear to be well done, but I have a few observations.

Major comments

-It is unclear to me why the authors used only those three time points. It is also unclear to what exactly they compare to *Dictyostelium* (same time points? similar?). This should all be better explained.

-I wonder if the authors could also compared the data to the transcriptomic data from another multicellular aggregate, that of *Capsaspora owczarzaki* (published in <https://doi.org/10.7554/eLife.01287>). That could be useful for the interpretations of the general pathways.

Minor comments

-line 31- "In the only other AGM model". There is *C. owczarzaki* and *Fonticula alba* as the authors discuss later on.

-line 39. "much of this similarity is shared with the clonal multicellularity of animals". I do not see that on the paper. Moreover, there is no comparison with aggregation in animals, so I do not think this can be said.

-line 525. "This is should". Replace to "This should".

Version 1:

Reviewer comments:

Reviewer #1

(Remarks to the Author)

Overall, the authors did a commendable job responding to reviewer concerns. We have noted a few items that they may wish to consider as they prepare their manuscript for publication:

1. The statement that *N. fowleri* is a parasite with a reduced genome and nearly half the number of genes as *N. gruberi* is not correct. *N. fowleri* is an opportunistic pathogen (not an obligate parasite), and has similar numbers of genes to *N. gruberi* (for example, see <https://bmcbiol.biomedcentral.com/articles/10.1186/s12915-021-01078-1/tables/2>).

2. Line 269: the authors refer to a "universal" autophagosome. In the rebuttal they justify the use of the term "universal" because "There may be additional or auxiliary autophagic proteins in other organisms, in fact there undoubtedly are taxon-specific variations, but these would not be universal if they are absent from Amorphea." This seems to put Amorphea in a taxonomically privileged position, such that genes absent from Amorphea are "auxiliary" and therefore not part of the "universal" autophagosome. This precludes the possibility that Amorphea has lost components of a putatively "universal" autophagosome. Yet it seems reasonable that otherwise well conserved autophagosomal genes may also have been lost from the Amorphea lineage. Without also performing an analysis of autophagosome components starting with baits selected from autophagic organisms from diverse taxa (e.g. plants or algae), the use of "universal" does not seem warranted. Therefore, it seems that the term "consensus" would be a better descriptor.

3. Lines 111-112: The authors state that 88% of membrane proteins are multicopy and reference Table S6 but it is not clear how to verify this claim from that table. Can the membrane-associated OGs be marked in the Table?

4. Lines 167-168: "Aggregation in *Acrasis kona* shows substantial increased expression of only 448 accessions (SDE Aggup), some of which are also among the 453 SDE Aggdn accessions (Table 3A, S11)." It does not seem logically possible for a gene to be both Aggup and Aggdn, and we wondered if this is a typo and that the authors must mean that many Aggup genes are also Germup/Germdn? We confirmed that there is no overlap in the lists of Aggup and Aggdn genes in Table S11. This should be clarified. A related point, in line 169 the authors mention "even without excluding these redundancies the Aggup+Aggdn total still accounts for less than 6% of the predicted proteome." It is not clear what redundancies the authors are referring to, since there are no redundancies in the list of Aggup and Aggdn genes.

5. Lines 225-227: the authors make a claim about proportions of genes ("including similar proportions of accessions in nearly all functional categories"), but the Figure 5A shows the absolute number of genes, not proportions. It would be helpful to the reader if the authors indicate the total number of homologs under consideration in the Ako 0h and Ddi 0h groups, either in the Figure legend or on the graph (perhaps under the Ako 0h and Ddi 0h text?).

6. Line 321: Note that moderate RPKM of kinetochore genes could also be due to high expression in a subset of cells, if they are dividing asynchronously.

Typos

295: Figure 7 erroneously referenced to describe loss of expression in germinating spores

278: ref should be Figure 6?

294: A. should be *A. kona*.

297: Figure S6 should be S5.

305: Figure 6 → Figure S5?

Figure S6: Legend is cut off, including the description of the blue cells – presumably indicating genes that were used as queries?

341 - 392: 341: references to Figure 7 should be changed to Figure 8.

442: Agg- → Agg-specs?

448: Unclear what Figure 4A refers to here. Figure 2?

Figure 6: evidene → evidence (this was already pointed out in the initial review!)

Reviewer #2

(Remarks to the Author)

Thanks to the authors for the revised version of the manuscript. I was already happy with the previous version. I believe they adequately address the points addressed.

RESPONSE TO REVIEWERS' COMMENTS

Reviewer #1 (Remarks to the Author):

Sheikh, Fu, et al. present the genome of *Acrasis kona*, an amoeba in the evolutionarily important and understudied Heterolobosean phylum of eukaryotes. Acrasids have a remarkable life cycle that includes aggregative multicellularity and formation of a sorocarp, currently the only organisms in this part of the eukaryotic tree known to manifest these behaviors. The authors conduct in-depth analysis of the *A. kona* genomic repertoire, combined with transcriptomic analysis of major stages during transition to the multicellular stage. Comparing the multicellular transition of *A. kona* to that of *Dictyostelium discoideum*, the classic model of aggregative multicellularity, the authors find numerous differences in gene expression patterns, while identifying potentially shared signaling motifs between these two divergent organisms.

The genomic resources presented here are important and will certainly spur the field in unraveling the many cell biological mysteries remaining in this understudied eukaryotic group. The choice of this particular species for in-depth genomic analysis is all the more appropriate due to the intense recent interest in the evolutionary origins of eukaryotic multicellularity, for which *A. kona* currently provides a taxonomically distinct example from existing model systems with sequenced genomes. This work will be of significant interest for researchers in comparative and evolutionary cell biology who are tracing the development of multicellularity. Furthermore, the addition of another well-annotated genome in the Heterolobosea will also enable further studies of the nature of the last eukaryotic common ancestor. The paper is well written and reflects the breadth of questions that will now be accessible with the publication of this genome.

Unfortunately, although the authors have deposited the genome scaffolds in NCBI, the gene models have not been deposited, making confirmation of the reported findings impossible. Moreover, the poor presentation among several of the primary figures obscures many of the major contributions of the work and—more importantly—further impedes verification of many of the authors' claims. Resolving these issues will require depositing the gene models in NCBI, re-analysis of existing data, significant changes to the presentation of figures and underlying data, as well as textual changes, as outlined below:

Major points

1. In order for other scientists to assess the work, confirm findings, and build on these analyses, the authors must make the gene models publicly available. Currently, the NCBI accession provided only includes genome scaffolds and the RNAseq data sets, not the gene models. The "Accession IDs" associated with predicted protein sequences included in the supplement do not link to the gene models. The "Accession IDs" associated with predicted protein sequences included in the supplement do not link to anything we can find in NCBI.

*We apologize for this, which was due to a protracted struggle with GenBank. The full set of *A. kona* gene models are now publicly available at DDBJ/ENA/GenBank under the accession JAOPGA000000000 (version JAOPGA020000000). The individual gene models also retain their original accession numbers (with "AK_" replaced with "AKO1_0") so that none of the original analyses needed to be repeated unless requested by reviewers. We also provide a*

fasta file of the full predicted proteome as a supplemental data set to accompany the manuscript.

2. Our biggest concern with the analyses as presented (and as reviewable without access to the gene models) is with the validity of the claim that relatively few genes are differentially expressed during *A. kona* aggregation. As far as we can tell, this conclusion is based on RNA-Seq with only one replicate per condition. While the authors take care to use the most appropriate statistical tool for this scenario (GFOLD), nevertheless the lack of replicates necessarily results in more conservative differential expression calls.

In comparing the number of differentially expressed genes in *A. kona* to a dataset from *D. discoideum* with multiple replicates, one might expect to find more differentially expressed genes in *D. discoideum* simply due to the greater number of replicates providing more statistical power. The methods suggest that the authors did not perform any comparable analysis on the Dicty data to the Acrasis analysis: specifically, how many genes are differentially expressed for Dicty when considering only one replicate from the Dicty data, passed through GFOLD as with *A. kona*?

This apples-to-apples analysis of RNA-Seq data run through the same statistical pipeline is critical for comparing differential expression between these two groups. This will require downloading the Dicty raw data from GEO and assessing only one replicate at a time using GFOLD.

At the reviewer's request, we have re-analyzed the Ddi AX4 RNaseq data separately for two replicates using Gfold, although for consistency we use the data from Santhanam et al. (2015) rather than GEO. These Gfold numbers are lower for Aggup genes (1859-2078 vs 2712) but higher for Aggdn (2069-2375 vs 1898). Thus, aggregating Ddi AX4 could be inducing only 4-fold as many genes as A. kona, rather than 5-fold. This is a useful point, but does not essentially change our conclusion, i.e., that Ddi AX4 shows a much more extensive aggregation response than A. kona.

We would also emphasize again that the number of A. kona Aggup genes is strikingly similar to the 500 "developmentally essential genes" identified in Dictyostelium, when development is induced in non-starving cells using rapamycin (Jaiswal and Kimmel 2019).

Changes:

GFold values for Ddi AX4 have been added to Table 3 (formerly Figure 4A) and noted in the text (lines 167-168). We also acknowledge that the lack of replicates and use of Gfold to calculate DE for A. kona allows for the possibility of up to ~25% more A. kona Aggup genes, although the bulk of these would mostly likely be lowly expressed genes (xxxx).

3. A *related* point: the fact that only 1 replicate was used per condition is extremely important for the reader to appreciate the data, and this should be stated more prominently by explicitly indicating how many replicates were performed for each condition (currently the statement in line 676-677 "as replicates were unavailable for each stage" is ambiguous as to if replicates were available for SOME stages but not all).

We now point out the single-replicate nature of our data in Results,

- *Section: Developmental gene expression in Acrasis kona (lines 159-160)*
- *Section: Development in Acrasis kona versus Dictyostelium discoideum (lines 167-168)*

The statement in Methods has also been changed to:

- *“As only a single RNAseq replicate was available for each stage, GFOLD (v1.1.4) (Feng et al. 2012) was used to calculate fold change.” (lines 731-733)*

4. The authors should also explicitly acknowledge the **alternative hypothesis** to many of their claims that that aggregation in *A. kona* (including genes like autophagy genes) is **regulated post-transcriptionally**, in contrast to *D. discoideum*.

We don't understand this comment. We have no evidence of post-transcriptional gene regulation, and we make no claims to this effect. For example, how could autophagy genes be post-transcriptionally regulated if they are not expressed?

*We would also further emphasize that *A. kona* Aggup number is strikingly close to non-starving development in *Dictyostelium* (Jaiswal & Kimmel 2019)*

5. The autophagy section is based on the **assumption that Acrasis and related species undergo autophagy using the same genes/pathways as Opisthokonts and Dictyostelia**. This is a major assumption and should be stated as such up front.

It is always possible that a strongly sequence-conserved gene has a non-canonical function or dual functions in other taxa. However, this is unlikely for a full suite of largely sequence-conserved genes covering a complex pathway such as autophagy. Therefore, we don't consider it a major assumption that autophagy functions similarly in Acrasis as in other eukaryotes. Although Discoba are distant relatives of Amorphea, they are still fully modern and metabolically-conservative eukaryotes (Fritz-Leyland et al. 2010).

*Furthermore, autophagy is well-documented in the discobid *Trypanosoma*, where it appears to function similarly to autophagy in other eukaryotes (reviewed by Brennand et al. 2012). In fact, morphological evidence of autophagy in *Trypanosoma* was first observed in the 1970s (Vickerman et al. 1977). Although the trypanosomatid autophagy genes are generally poorly sequence conserved, this is a common phenomenon in obligate parasites, even in the most basic informational pathways. Therefore, it is not so surprising that we are able to identify more autophagy homologs in Acrasis, and many of these are highly sequence-conserved in the amoeba.*

Changes:

We now cite Brennand et al. (2012) as well as Kuo et al. (2018) regarding the antiquity of autophagy among eukaryotes and high divergence of kinetoplastid sequences (lines 258-263).

6. The HGT analyses appear to be based only on comparisons to predicted proteins from gene models. Gene prediction in these species is notoriously difficult, and the *Naegleria gruberi* genome is known to have errors in gene prediction. The authors should use tBLASTn or similar to search for the putative HGTs on scaffolds of *Naegleria* and other Discobans before asserting that these are “confirmed HGTs”.

*Evidence that the *N. gruberi* genome annotation is not grossly incomplete and potentially missing a substantial number of the *A. kona* HGT candidates:*

- *Gene numbers and BUSCO scores for the *Naegleria* genomes in general, and *N. gruberi* in particular, are high and compare well with other protists (table below)*

- There is no indication that *N. gruberi* uses an alternative genetic code or abundant non-canonical intron splice sites, and the genome is only mildly GC-rich (57%).
- HGT genes are especially unlikely to escape detection during annotation since, by definition, they have homologs in other species.

	Genome	Id'd genes	BUSCO	
N. gruberi	42 mB	15,717	85%	Table S2
N. lovaniensis	30.8 mB	11,305	84%	Joseph et al 2021
N. fowleri	27.9 mB	~9,500	87%	“
Ddi AX4	34 mB	12,500		Eichinger et al. 2005
Aureococcus	57-73 mB	13-17,000		Gann et al. 2021
Volvox carteri	138 mB	14,520		

Therefore, it is unlikely that a substantial number of *N. gruberi* genes have escaped detection, certainly not enough to account for the 3-fold greater HGT candidates we find in *A. kona*. In order for this to be an annotation artefact, it would require 60% of the *N. gruberi* genes to be missing. Including additional *Discoba* in our analyses would also not likely make much difference, as these are almost exclusively transcriptome data except for the highly reduced and sequence-divergent genomes of the kinetoplastid parasites (eg *Trypanosoma*).

We have striven to be conservative in our interpretation of the *A. kona* HGT data, limiting our analyses to HGTs with wide-taxonomic distribution and at least moderately strong phylogenetic support (Figure 2). This is despite the fact that the numerous uniHits are also likely to be largely legitimate HGTs (eg Figure 3). We also try to avoid making quantitative statements, but rather focus on general trends. Thus, we feel the results are sufficient to assert that the general volume of HGT in the acrasid lineage appears to be large and the diversity of likely donors is broad but markedly biased toward a few distinct taxa.

Changes:

Given the general lack of genome data from *Discoba* or other excavates, we have tried to emphasize the preliminary nature of our findings. To this end, we have replaced cHGT (“confirmed HGT”) with HGT \geq 60 (“HGT with \geq 60% bootstrap support”), which is more precise and conservative. We also now include figures for all the *Aggup* HGT \geq 60s, which utilize BLASTp against the full GenBank nr database (Figures S4a-S4ag, Table S13).

We have also modified the wording in several places to point out the general lack of data for *Discoba* and possible impact of unannotated genes in various genomes (lines).

7. The graphical abstract demonstrates the *A. kona* cytoskeleton using cytoplasmic microtubules and a cytoplasmic centrosome (including centrioles), yet neither of these has been observed in *A. kona*, and in fact there is some evidence that this organism may lack cytoplasmic microtubules like the related *Naegleria gruberi* (Roos UP and Guhl B. (1996) *Eur J Protistol* 32: 171-189). We propose that the authors choose a more accurate way to demonstrate the cytoskeleton by illustrating actin instead, since it is known to be abundant in *A. kona* cytoplasm (Hellstén M and Roos UP. (1998). *Fungal Genet Biol* 24: 123-145). Alternatively if the authors insist on using microtubules, then they should use the one currently documented microtubule structure in *A. kona*, the intra-nuclear spindle (Roos and Guhl, *vide supra*).

Thank you for this.. Since Roos and Guhl only observed microtubules during mitosis and our figure is meant to depict an aggregating and therefore non-dividing cell, we have deleted the microtubules from the figure and replaced them with a schematic actin cytoskeleton.

8. The utility of the pie charts in Figure S1 is unclear; it seems a well laid out table would be a more useful way for the reader to make comparisons (rather than scrunching that into the key).

Changes:

Thank you for this suggestion. We have converted this to a table (new Table 2) and placed it in the main body of the text.

9. The data underlying Figure S2 are not available, and the figure itself is missing a lot of information, such as labels for different metabolic steps.

We have only labelled the major pathways because any finer detail would be unreadable.

Changes:

We have added a new supplemental data table (Table S8), which lists all of the A. kona metabolic pathways and their corresponding Expasy map reference. The table is also noted in the legend to the figure (now Figure S1) along with the Expasy URL.

10. The data tables are difficult to understand. Specifically, we were not able to use Figure 4 to verify a number of claims made by the authors:

◦ Figure 4B is extremely difficult to interpret. 3D bar graphs are very troublesome for the reader to interpret in general, but because the bars on this graph extend below zero on the y-axis, it is in this case illegible and impossible to understand. The authors should re-create this graph using grouped bar charts in a conventional 2D presentation, or consider using UpSet plots (see below).

Changes: We have split the figure as follows.

- *Figure 4A is now Table 3. It has also been reorganized so that all sections sum to the same value for each column (see below).*
- *Figure 4B (now 4A and 4B) consists of two parts. Part A shows time point 0h (actively growing cells), and Part B shows time point 5h (aggregating cells). This obviates the need for a 3D graph, and we agree it is easier to read and interpret. The categories and numbers of accessions for each category for each taxon and time point are taken directly from Table S11, as stated in the figure legend.*

◦ Many assertions are based on adding or dividing different columns in figure 4A, which is a tiresome exercise for the reader, and in many cases we cannot verify the numbers in the text. Please re-do this table to make your points clear, and follow a standard convention that if there is a row in the table for “all” or “total”, then it should be clear which other rows in the table will sum to the “all” row. Some specific issues:

Although most of the categories are binary, we agree that it adds clarity to include the reverse of each category so that columns sum to the same total for each section.

Changes:

The Table 3 (formerly Figure 4A) is now divided into three sections, and each section sums to the same total for each column (life cycle transition)..

- For Aggup/Aggdn/Germup/Germdn, the number of accessions in ‘novel,’ ‘Multicopy,’ ‘cHGT,’ and ‘TM +/- SignalP’ don’t add up to ‘all.’ While we acknowledge that these categories don’t seem to be mutually exclusive, in that case we would then expect the sum to add up to more than the ‘all’ row due to double-counting. Instead, sometimes the sum adds up to less than the ‘all’ row (e.g. for Aggup), suggesting some genes in the ‘all’ category are not broken down! This is confusing. We encourage the use of horizontal lines to separate logically distinct groupings of mutually exclusive categories, and ideally within each grouping the sum of accessions should add up to ‘all’ at the top (Though perhaps an exception to this rule could be made for simple binary categories like ‘has TM’ and ‘does not have TM’).

All columns now sum to the same total for each section (see above).

- Line 157 states that Germup and Germdn show change of 1771 and 2033 accessions, respectively. We are unable to reproduce these numbers using the data in the table.

Changes:

We have added a new supplemental data table (Table S11) listing all Aggup/dn and Germup/dn accessions along with their respective DE and RPKM values.

- Lines 164-165 mention 1214 OGs involved in growth and references Figure 4A. But Figure 4A shows differences between Agg, Gro, Germ, not Gro explicitly, so this claim cannot be verified from the table (is Aggdn + Germup taken to be synonymous with growth? If so, we still cannot get to 1214 OGs).

This is the total of Germup + Germdn, i.e., the number of accessions substantially affected by germination. This is similar to the claims made for the Ddi AX4 developmental transcriptome, i.e. that it affects 1/3 of the genome (Aggup + Aggdn; e.g., Parikh et al 2010). However, we agree it has limited utility here.

Changes:

Rather than analyse this in the detail needed to make more sense of it, we have deleted all reference to non-developmental OG expression.

- In general the claims related to Figure 4A and 4B would benefit from an UpSet plot (doi:10.1109/TVCG.2014.2346248, can be made using the UpSetR package in R), which will allow the author to break the gene categories down into whichever granularity they choose and allow explicit comparison between claims in the text and specific bars in the table. For example, Line 205-207 describes the overlap between two categories: “Aggup” and “AKo 0 h RPKM > 10” based on the fact that bar charts are shown for both of these. But the overlap between these categories is not shown in Figure 4B, so this claim cannot actually be verified based on this data!

We have chosen instead to convert the figure to a table (Table 3, formerly Figure 4A) and then split the bar chart into separate time points (Figures 4A and 4B, formerly Figure 4B). This makes 4A and 4B very simple figures, which should be easy to read and interpret. The most important comparison here is between the two amoebae at individual time points, rather than between time points. If the reader is interested in greater detail, this can be read directly off Table S12, which is organized by the same categories as the figure. Defining protein function in finer detail would risk over-interpretation, since protein function is based on imputed homology, and proteins maybe multi-functional.

11. Beyond Figure 4 there are a number of claims that cannot be verified from the information provided. More information should be provided or claims should be adjusted. LSEP

◦ Line 175 states that “ none of the 35 A. kona Aggup cHGTs trace to Dictyostelia based on BLASTp or phylogeny.” First, we count 36 Aggup cHGTs in Table S11. But more importantly we are not sure how to evaluate this claim. For example, from just a cursory inspection it seems that at least 2 genes in Table S11 with a cHGT value > 60% have a strong BLASTp hit in Dictyostelium (AKO1_01251, nfyA, top hit in Dicty has e-value 9e-21 to a gene annotated nfyA; AKO1_14998 clp1, top hit in Dicty has e-value 7e-108). Are these genes ruled out by phylogeny? If so then how do we evaluate the phylogenetic screening (it does not seem to be shown in Table S11)? If the authors want to make a claim based on Table S11, it should be more explicit in the table, or the claim needs to be broken down into verifiable pieces.

BLAST e-value is not a reliable indicator of phylogeny, which is why all accessions with non-excavate top hits were tested by phylogenetic analysis where feasible (i.e., all multiHits). The resulting trees Aggup HGT≥60 have been added to Supplemental data as follows:

Changes:

- *Phylogenies for the 35 Aggup HGT≤60 accessions are shown in Figure S4a-Sag.*
- *A new supplemental table (Table S13) lists the 35 accessions and details of their phylogenies - alignment length, phylogenetic method and model, tree description in Newick format.*
- *The Aggup HGT accessions are annotated in Table S12 with their tree figure number (column K).*

◦ Line 348: how are we to understand analogy and homology from this diagram? The external signaling pathways are scattered across all the categories – it is very difficult to assess this claim. Perhaps the authors could mark which genes they are defining as pertaining to external signals to help the reader follow this (extremely busy and confusing) figure?

All Acrasis accessions are assigned function based on BLASTp, i.e., based on homology. Full details of data supporting assignment of homology are given in Tables S12 and S19. The only non-homologs are the predicted membrane receptors, which of course are novel.

Changes:

- *We have broken the figure down into two parts, both of which are now enlarged separately and should be easier to read.*
- *Part A is an enlarged and slightly streamlined version of the diagram.*

- *Part B is the tables, also enlarged, but still color-coded to match the diagram. The sections are organized by pathway, as this seemed most logical, with a separate section for novel predicted membrane receptors.*

◦ Line 381-382: “13% of the aggregating cell protein production is potentially involved in ECM construction” This claim does not follow from the data – protein production is not measured. The authors could instead comment on the diversity of expressed genes or something like that.

The wording has been changed as follows:

- *“...six of the 62 potential ECM proteins are highly expressed (RPKM >1000) and 18 altogether are expressed with RPKM >100 (Table S20). This suggests the possibility that a substantial portion of A. kona protein production during aggregation could be devoted to constructing the acrasid’s ECM.” (lines 414-417).*

◦ Line 395: where do these numbers come from? We can’t figure out how to get them from the table shown.

This is the sum of entries (individual OGs) in column A of (former) Table S19. However, the corresponding table and section of the manuscript (Multigene families) is now deleted. We felt that this large and complicated table was better presented elsewhere, where it could be dealt with in more detail.

Minor points:

- Can the authors make any comments on the possible ploidy of A. kona, perhaps based on the proportions of SNVs in their sequencing data?

Not at this time.

- Confused by Agg vs Germ and Gro vs Agg nomenclature: which is denominator? Aggup and Aggdn were clearer, maybe just explicitly state that Agg vs Germ = Agg / Germ or something like that.

Agg v Germ refers to all changes between aggregation and germination, therefore both Germup and Germdn. Likewise, Gro vs Agg refers to both Aggup and Aggdn. This should be clear from the headings in Table 3, which should be easier to read now as a separate table. We have also added this further clarification to the Table 3 legend...

- *“Numbers of substantially differentially expressed accessions (accs) are shown for the transition from asocial feeding to social development (growth to aggregation - Agg vs Gro: Aggup/Aggdn) and from development back to asocial growth (aggregation to germination - Agg vs Germ: Germup/ Germdn) (Table S11)”*

- Figure 3: How did the authors decide to focus on this particular list of signaling domains?

The list was taken from Fritz-Leyland et al. 2010. This is now explicitly stated in the figure legend, rather than simply acknowledged.

- Line 99: N. fowleri also have a well-characterized genome at this point (2 papers, fairly extensive)? The authors should amend this point to indicate “one of only two members of the

phylum Heterlobosea with a well-characterized genome (the other being the closely related and pathogenic *N. fowleri*)”.

N. fowleri is a parasite and has a greatly reduced genome relative to N. gruberi, with nearly half as many predicted protein genes and probably missing many metabolic pathways. This limits its utility for comparative genomics.

Changes:

We have changed the wording to “the only other non-parasitic Heterolobosean with a well-characterised genome” (line 103).

- Line 128-129 - It is a stretch to say that the HGT data “confirms” that acrasids are omnivorous micro-predators that “probably” acquire HGTs from prey. The authors should walk these back along the lines of: “consistent with being omnivorous” and “may acquire HGTs from prey.”

Done.

- Line 169-170: This claim would be strengthened by some quantitative or statistical argument supporting the use of the term “enriched;” a chi-square or Fisher exact test for the association between genetic redundancy/novelty vs. agg/growth/germ would be useful.

The statement has been deleted.

- Line 239: The authors construct a “universal” autophagosome based only on species from Amorphea... We suggest choosing a different word that is more reflective of the (necessarily) limited taxonomic sampling underlying this categorization. What about “consensus”?

The Amorphea sequences are used only to identify candidate queries. The results of the analyses using these queries is what defines a eukaryote-wide autophagosome. There may be additional or auxiliary autophagic proteins in other organisms, in fact there undoubtedly are taxon-specific variations, but these would not be universal if they are absent from Amorphea.

- Line 275-277: Why does this appear unlikely? There is no evidence for discarding this hypothesis after spending a whole paragraph discussing it.

The statement has been changed to: “we see no evidence of increased exosome production, which could provide additional nucleotide building blocks during starvation in either amoeba” (lines 106-108).

- Line 305: Is there any evidence that stalk cysts can de-differentiate into trophozoites? This seems relevant to the speculation in the Discussion about the fate of stalk cysts after spore dispersal (lines 468-469)

If the reviewer is referring to germination as de-differentiation, then the answer is yes, although this would be an usual usage of the term. Both stalk cysts and spores are viable and germinate to produce live amoebae. This was shown in the very earliest description of the species (cf Olive 1975).

- Line 449: strong claim based on this data, especially since there is not any functional evidence linking any of these genes with phenotypes. The authors should soften this language.

“probably” has been replaced with “could have been”.

- Line 453: We understand that “starvation” here means that food has been depleted, but this seems to undercut the notion that the A. kona cells are not starving while the D. discoideum cells are. Perhaps a different word would help disambiguate the two senses of ‘starvation?’

The statement is correct - both amoebae can be induced to develop in the lab by starvation. However they can also be induced to develop in the lab by other means, such as dense culture and/or blue light (Acrasis) or inhibition of mTORC1 by rapamycin (Dictyostelium).

The wording has been modified from “are induced” to “are commonly induced” to avoid any possible confusion (line 523).

- Line 589: “the acrasid appears to be...” is too strong in the absence of functional data for any of these genes. To my ear “appears to be” signals an evident conclusion, for example in the previous sentence where you use “appears to be” to describe an observation from the data. Weaken this to “the acrasid may be...”

Done.

Typos

- Line 162: Table 4A → Figure 4A
- Line 445: Table 11 → Table S11
- Figure 4 legend, last sentence: “Differentially expression” → “Differential expression”
- Title of Figure S8 should be related to kinetochore or cell cycle not autophagy
- Figure 5: “evidene” should be “evidence”

Reviewer #2 (Remarks to the Author):

The authors report the genome sequence of the protist Acrasis Kona, a member of the Excavata clade. They also present transcriptomic sequences from three different moments of the life cycle, which includes an aggregative multicellular stage with fruiting bodies. These include growth, aggregation, and spore germination. The authors analyze the genes significantly upregulated in those time points and compare their function to those found in Dictyostelium, which also develops into multicellular fruiting bodies. The authors suggest that A. kona uses very similar pathways on its development as Dictyostelium.

The community will be very pleased with the genome and transcriptomic data here reported. The analyses appear to be well done, but I have a few observations.

Major comments

-It is unclear to me why the authors used only those three time points. It is also unclear to what exactly they compare to *Dictyostelium* (same time points? similar?). This should all be better explained.

The time points correspond to before, during and after development. We have expanded the explanation the time points as follows (lines 152-154):

- “To gain insight into *Acrasis kona* development, we sequenced transcriptomes from pre-, mid- and post-developmental cells, which correspond to the primary life cycle stages of growth (Gro), aggregation (Agg) and spore germination (Germ), respectively (Figure 1)”.

*The corresponding time points in *Dictyostelium* are well-established: 0h starvation = actively growing cells and 5h starvation = mid-aggregation (Santhanam et al. 2015).*

*Correlation of starvation time points in *Dicty* vs *Agg* in *Acrasis* is shown in Figure S2.*

-I wonder if the authors could also compared the data to the transcriptomic data from another multicellular aggregate, that of *Capsaspora owczarzaki* (published in <https://doi.org/10.7554/eLife.01287>). That could be useful for the interpretations of the general pathways.

*Gene-specific expression data on *Capsaspora* aggregation is not publicly available except as raw unassembled SRA files. Published analyses provide only numbers of genes in broad GO categories; we could not find even partial lists of *Capsaspora* aggregation genes in any of the supplemental data. The main focus of the above publication is unique similarities between *Capsaspora* and Metazoa, which of course are not useful here.*

*Aggregation in *Capsaspora* is also a transient and reversible state, rather than a life cycle stage, and its function in the organism is unknown. It also does not seem to involve differentiation, fruiting body construction, or spore production. This is very different from aggregation in *Dictyostelium*, which performs the same general function as in *Acrasis*, i.e., production and dispersal of spores.*

*In sum, a comparison of aggregation in *Acrasis* and *Capsaspora* is not feasible at this time, short of our reanalysing the *Capsaspora* SRA files ourselves, and, given that we are already analysing a member of supraprokingdom Amorphea (*Dictyostelium*), it is unclear what more such a comparison would contribute.*

Minor comments

-line 31- “In the only other AGM model”. There is *C. owczarzaki* and *Fonticula alba* as the authors discuss later on. -line 31- “In the only other AGM model”. There is *C. owczarzaki* and *Fonticula alba* as the authors discuss later on.

*We have modified the wording to avoid any possible offense. However, it must be acknowledged that developmental data for *Capsaspora* and *Fonticula* are limited and not easily accessible, with the exception of unassembled SRA sequence files. *Dictyostelium*, on the other hand is supported by a large body of easily accessible data. These include fully*

assembled RNAseq data for multiple time points (e.g. Santhanam et al. 2015) and DictyBase with bioinformatic and functional annotation of all genes, mutants and mutant combinations, GO annotation, supporting publications and more.

Changes:

We now refer to Dicty as the only “extensively studied” AGM model (lines 55-57).

-line 39. “much of this similarity is shared with the clonal multicellularity of animals”. I do not see that on the paper. Moreover, there is no comparison with aggregation in animals, so I do not think this can be said.

The sentence does not exclude aggregation in animals, so it is correct as written. We have modified the sentence in order to try to make it clearer, as well as more specific.

Changes:

- *Highly conserved homologs of many of these proteins also play roles in similar pathways in the clonal development of animals. (lines 39-40).*

-line 525. “This is should”. Replace to “This should”.

Done.

RESPONSE TO REVIEWERS' COMMENTS

Reviewer #1 (Remarks to the Author):

Overall, the authors did a commendable job responding to reviewer concerns. We have noted a few items that they may wish to consider as they prepare their manuscript for publication:

1. The statement that *N. fowleri* is a parasite with a reduced genome and nearly half the number of genes as *N. gruberi* is not correct. *N. fowleri* is an opportunistic pathogen (not an obligate parasite), and has similar numbers of genes to *N. gruberi* (for example, see <https://bmcbiol.biomedcentral.com/articles/10.1186/s12915-021-01078-1/tables/2>).

We were relying on earlier reports, which apparently were incorrect. All mention of N. fowleri is now deleted.

2. Line 269: the authors refer to a “universal” autophagosome. In the rebuttal they justify the use of the term “universal” because “There may be additional or auxiliary autophagic proteins in other organisms, in fact there undoubtedly are taxon-specific variations, but these would not be universal if they are absent from Amorphea.” This seems to put Amorphea in a taxonomically privileged position, such that genes absent from Amorphea are “auxiliary” and therefore not part of the “universal” autophagosome. This precludes the possibility that Amorphea has lost components of a putatively “universal” autophagosome. Yet it seems reasonable that otherwise well conserved autophagosomal genes may also have been lost from the Amorphea lineage. Without also performing an analysis of autophagosome components starting with baits selected from autophagic organisms from diverse taxa (e.g. plants or algae), the use of “universal” does not seem warranted. Therefore, it seems that the term “consensus” would be a better descriptor.

There are two main points here. 1) This is the first analysis of autophagy to include a non-parasitic Discobid, which are an outgroup to both Amorphea and Diaphoretickes (plants and algae). Thus we extend the analysis of autophagy to much deeper in the eukaryote tree. 2) Previous comparative work on autophagy is problematic because of the emphasis on Saccharomyces, which has a very divergent autophagosome both in terms of sequence and gene content. Therefore, we had to start from scratch to identify probes. We also surveyed across a far wider diversity of eukaryotes (including plants and all major groups of algae) than previously done. This makes our work the broadest treatment to date of autophagosome evolution and without some of the biases of previous work. Our use of the term “universal” was meant to emphasize this.

However, we are well aware that there are even deeper branches of eukaryotes that we have not included (the former Metamonada, Al Jewari and Baldauf 2023). Therefore we are willing to abandon “universal”. However, we think consensus would probably be equally or even more misleading. We now refer to Figure 6 simply as “The eukaryotic autophagosome”.

One final point - while we appreciate the value of experimental evidence, sequence similarity in itself is a powerful tool for discovering homology. Homology, in turn is evidence of shared function, especially when this homology extends to strong sequence similarity for a substantial number of interacting proteins, in this case nearly the complete human autophagosome.

Action:

We have substituted “eukaryotic” for “universal predicted autophagosome” and further clarify the reasoning behind our approach (lines 266-269).

3. Lines 111-112: The authors state that 88% of membrane proteins are multicopy and reference Table S6 but it is not clear how to verify this claim from that table. Can the membrane-associated OGs be marked in the Table?

We have now annotated all TM proteins belonging to OGs with the relevant OG ID numbers in Supplemental Data file 1. We also correct and clarify this typo as follows: “63% of proteins with four or more transmembrane domains are assigned to orthology groups” (lines 105-107).

4. Lines 167-168: “Aggregation in *Acrasis kona* shows substantial increased expression of only 448 accessions (SDE Aggup), some of which are also among the 453 SDE Aggdn accessions (Table 3A, S11).” It does not seem logically possible for a gene to be both Aggup and Aggdn, and we wondered if this is a typo and that the authors must mean that many Aggup genes are also Germup/Germdn?

We confirmed that there is no overlap in the lists of Aggup and Aggdn genes in Table S11. This should be clarified. A related point, in line 169 the authors mention “even without excluding these redundancies the Aggup+Aggdn total still accounts for less than 6% of the predicted proteome.” It is not clear what redundancies the authors are referring to, since there are no redundancies in the list of Aggup and Aggdn genes.

Yes, this was a misunderstanding and has been corrected.

5. Lines 225-227: the authors make a claim about proportions of genes (“including similar proportions of accessions in nearly all functional categories”), but the Figure 5A shows the absolute number of genes, not proportions. It would be helpful to the reader if the authors indicate the total number of homologs under consideration in the Ako 0h and Ddi 0h groups, either in the Figure legend or on the graph (perhaps under the Ako 0h and Ddi 0h text?).

The wording has been changed to “similar numbers”.

6. Line 321: Note that moderate RPKM of kinetochore genes could also be due to high expression in a subset of cells, if they are dividing asynchronously.

This is true. However, the fact remains that there is also no evidence of an increase in cell number from aggregation to sorocarp. Therefore, there is also no visible evidence that the cells are dividing.

Typos

295: Figure 7 erroneously referenced to describe loss of expression in germinating spores
Figure 7 is correct reference. We have further modified the wording to try to make this as clear as possible:

*... the entire set of **proteasomal protein genes** appears essentially shut down in germinating spores (Figure 7).... (lines 292-295).*

278: ref should be Figure 6?
corrected

294: A. should be A. kona.
corrected

297: Figure S6 should be S5.
corrected

305: Figure 6 → Figure S5?
corrected

Figure S6: Legend is cut off, including the description of the blue cells – presumably indicating genes that were used as queries?
corrected

341 - 392: 341: references to Figure 7 should be changed to Figure 8.
corrected

442: Agg- → Agg-specs?

corrected

448: Unclear what Figure 4A refers to here. Figure 2?

Changed to Table S5.

Figure 6: evidene → evidence (this was already pointed out in the initial review!)

corrected

Reviewer #2 (Remarks to the Author):

Thanks to the authors for the revised version of the manuscript. I was already happy with the previous version. I believe they adequately address the points addressed.